# The Impact of Diet and Physical Activity on Fat-to-Lean Mass Ratio

**DOI:** 10.3390/nu16010019

**Published:** 2023-12-20

**Authors:** Elvira Padua, Massimiliano Caprio, Alessandra Feraco, Elisabetta Camajani, Stefania Gorini, Andrea Armani, Bruno Ruscello, Alfonso Bellia, Rocky Strollo, Mauro Lombardo

**Affiliations:** 1Department of Human Sciences and Promotion of the Quality of Life, San Raffaele Open University, Via di Val Cannuta, 247, 00166 Rome, Italy; elvira.padua@uniroma5.it (E.P.); massimiliano.caprio@uniroma5.it (M.C.); alessandra.feraco@uniroma5.it (A.F.); elisabetta.camajani@uniroma5.it (E.C.); stefania.gorini@uniroma5.it (S.G.); andrea.armani@uniroma5.it (A.A.); bruno.ruscello@uniroma5.it (B.R.); rocky.strollo@uniroma5.it (R.S.); 2Laboratory of Cardiovascular Endocrinology, San Raffaele Research Institute, IRCCS San Raffaele Roma, Via di Val Cannuta, 247, 00166 Rome, Italy; 3Department of Systems Medicine, University of Rome “Tor Vergata”, Via Montpellier 1, 00133 Rome, Italy; bellia@med.uniroma2.it

**Keywords:** caloric restriction, diet mediterranean, body composition, exercise, energy metabolism

## Abstract

In this retrospective study, we evaluated the efficacy of a personalised low-calorie Mediterranean Diet (MD) in promoting fat mass (FM) reduction while preserving fat-free mass (FFM). This study involved 100 Caucasian adults aged 18–65 years who followed a tailored low-calorie MD for two months. The total energy expenditure was assessed using a multi-sensor armband. The change in body composition (BC) was evaluated using the Δ% FM-to-FFM ratio, calculated as the difference in the FM to FFM ratio before and after the diet, divided by the ratio before the diet, and multiplied by 100. A negative value indicates a greater decrease in FM than FFM, while a positive value suggests a greater increase in FM than FFM. This study demonstrated a significant FM reduction, with an average decrease of 5% (*p* < 0.001). However, the relationship between caloric reduction and the Δ% FM-to-FFM ratio showed a weak negative correlation (r = −0.03, *p* > 0.05). This suggests that the calorie deficit had a minimal direct impact on the BC changes. Subjects over the age of 30 showed an increase in muscle mass, while younger subjects showed no significant changes. Moreover, a direct correlation was observed between the changes in MET (Metabolic Equivalent of Task) values and the Δ% FM-to-FFM ratio, indicating that improved average physical activity intensity positively influences BC. In the female subgroup, high protein intake, exercise intensity, and the duration of physical activity were positively correlated with an improvement in the Δ% FM-to-FFM ratio. However, for individuals with BMI 20–25 kg/m^2^, high fibre intake was surprisingly negatively correlated with the Δ% FM-to-FFM ratio. This study underscores the intricate interplay between calorie restriction, physical activity intensity, and BC changes. It also suggests that individual factors, including age, gender, and BMI, may influence the response to a low-calorie MD. However, further prospective studies with larger sample sizes are necessary to confirm and expand upon these findings.

## 1. Introduction

Optimising body composition (BC) holds greater importance for long-term health than merely losing body weight (BW) [1]. In fact, fat mass (FM) serves as a more critical indicator of health risks, making it a more relevant focus than BW alone [2]. The goal in weight loss should be the reduction in FM, especially visceral fat, while minimising the loss of fat free mass (FFM) [3]. This objective becomes even more vital, considering that fluctuations in energy expenditure can not only affect the long-term success of weight loss but also contribute to weight regain [4].

Though adherence to the Mediterranean Diet (MD) is associated with lower obesity rates [5], evidence also suggests that reduced-calorie diets can result in meaningful BW loss, irrespective of macronutrient distribution [6]. The synergy of diet and exercise appears more effective in improving BC than either approach in isolation [7,8]. The degree of caloric restriction, type and amount of exercise, and the pace of weight loss are significant variables influencing the percentage of lost FFM [9]. Very low-calorie diets, for example, can offer fast results in FM loss but with an increased risk of FFM loss. Addressing the unclear optimal extent of calorie restriction for weight loss and BC is the primary focus of this study. Thus, this study will try to address the notable gap in research providing precise measurements of energy expenditure and sports parameters in a free-living context [10,11,12].

In the context of this study, we specifically focus on analysing the impact of calorie restriction within the context of the MD, aiming to reduce FM and simultaneously preserve FFM. This targeted approach allows for a deeper investigation of dietary dynamics and their influence on BC. Our hypothesis is that varying degrees of caloric restriction within the framework of the MD will have differential effects on reducing FM and preserving FFM. We base this hypothesis on the understanding that both the quality of the diet and caloric intake are pivotal in shaping BC outcomes. We suppose that a synergistic effect of diet and varied physical activities can more effectively influence the FM-to-FFM ratio.

## 2. Materials and Methods

In this retrospective study, we analysed a cohort of 100 Caucasian adults aged between 18 and 65 years, who had attended the specialised nutrition and metabolism medical centre in Rome, Italy, from January to March 2023. The clinical centre mainly serves people seeking clinical assessment and guidance regarding their BC and associated health problems. As a result, the study sample exhibited certain characteristics that may differ from the broader population in the country. Notably, participants in this study demonstrated a heightened awareness of and interest in BC, potentially influencing their dietary and physical activity patterns. Additionally, the medical centre’s specialized focus on BC assessment may have attracted individuals with specific health concerns or fitness goals. However, none of the participants were adhering to an MD or engaging in caloric restriction prior to the commencement of the study. Exclusion criteria were as follows: individuals who were pregnant or breastfeeding at the time of data collection, diabetics, patients on medications that affect weight (e.g., glucocorticoids, oestrogen, anticonvulsants), and those with specific medical conditions such as alcoholism or chronic kidney disease, as these conditions could impact metabolism. The patients were re-evaluated after two months as part of the clinic’s standard follow-up protocol. All the patients provided written informed consent, and the study design was approved by “Lazio area 5 territorial ethics committee”—(Approval Code: N.57/SR/23—Approval Date: 7 November 2023) in accordance with the Helsinki Declaration of 1964 (with later amendments). The subjects were assessed at two points: T0 (two months prior) and T1 (present time), allowing us to evaluate changes over the intervention period. Data were obtained from the medical records and integrated with the weekly food diaries kept by the participants. The combination of medical records and dietary diaries facilitated a comprehensive assessment of the effects of the MD and caloric restriction on participants’ BC over the duration of the study. To comprehensively evaluate the participants’ health status, biochemical parameters were assessed. This included glycemia, total cholesterol (TC), high-density lipoprotein (HDL), triglycerides (TG), systolic blood pressure (SBP), and diastolic blood pressure (DBP). These assessments were conducted as part of the initial health evaluation at T0.

### 2.1. Body Composition

All the participants went through a medical evaluation that included dietary history, physical examination, and BC assessment. Body measurements were taken after overnight fasting and while participants only wore underwear. The participants were weighed using calibrated electronic scales with a maximum weight limit of 250 kg. The scales were placed on a flat, uncarpeted surface and levelled. The participants, dressed in light clothing and without shoes, stood still on the scale while the clinic assistant recorded the weight to the nearest 100 g. Two readings were taken, and if they varied by more than 100 g, a third measurement was taken. The two closest measurements were used for analysis. For height measurements, a stadiometer was used and placed on an even, uncarpeted area. Participants removed their shoes and, if applicable, let down any hair tied atop their heads. They were positioned to face the clinic assistant, looking straight ahead with their head in the Frankfort plane. Their shoulders were relaxed, and they were instructed to stand in such a way that their shoulder blades, buttocks, and heels slightly touched the stadiometer’s stand. Their arms were relaxed at their sides, legs were straight, knees were together, and feet were flat with heels touching. Two height measurements were taken and recorded. If the measurements varied by more than 0.1 cm, a third measurement was taken, and the two closest measurements were used for further analysis. BC, including FM, FFM, and total body water (TBW), was assessed using the BIA Tanita BC-420 MA, a validated instrument compared to BodPod [13,14]. The device measures from a standing position without electrodes, with a precision of up to 100 g. Pre-assessment guidelines required participants to be at least three hours post-awakening and meals; 12 h post-strenuous exercise; and to avoid excessive eating, drinking, and alcohol 12 h prior. Women were advised to avoid measurement during menstruation.

FM was assessed in kilograms (kg) and as a percentage of total body weight (%). The FM-to-FFM ratio was calculated in kg. The change in this ratio over time, denoted as Δ% FM-to-FFM ratio, was = [(FM-to-FFM ratio at Time 1 (T1)—FM-to-FFM ratio at baseline (Time 0, T0))/FM-to-FFM ratio at T0] × 100. Δ% FM-to-FFM reflects the proportional change in FM relative to FFM during the study period. A negative Δ% FM-to-FFM indicates a reduction in FM compared to FFM, signifying a loss of fat while preserving muscle mass, which is beneficial for health. Conversely, a positive Δ% FM-to-FFM suggests an increase in FM relative to FFM.

### 2.2. Energy Expenditure

Participants wore the SenseWear Pro2 Armband (SWA, Bodymedia Inc.-manufactured in Pittsburgh, PA, USA) for a minimum of 48 h. The SWA is a compact, non-invasive device that combines accelerometery with heat-related sensors—such as galvanic skin response, heat flux, and skin and near-body temperature—to offer a more accurate estimate of energy expenditure. The device was calibrated according to manufacturer guidelines to ensure accuracy. The device incorporates a tri-axial accelerometer that measures movements along three axes, thereby improving the accuracy of physical activity and energy expenditure tracking [15,16]. Participants were instructed to maintain a consistent lifestyle and physical activity level throughout the study period. The armband recorded a range of data, including the duration of physical activity, time spent lying down, energy expenditure for activities with an intensity greater than 3 Metabolic Equivalents (METs), number of steps taken, hours of sleep, daily METs, daily wear time, and total energy expenditure (TEE). Subjects wore the armband only at T0. Data from the SWA were downloaded and analysed by trained research personnel. The variable Δ Kcal% indicates the difference between the total energy expenditure measured with the armband and the caloric intake. Δ Kcal% = 100 − (Energy (diet)/TEE) × 100 Both values refer to T0.

### 2.3. Diet

At T0, an individualised MD was prepared for the patients based on their previous habits. Additionally, specific measurements are provided for protein, fibre, saturated fatty acids (SFA), polyunsaturated fatty acids (PUFA), and monounsaturated fatty acids (MUFA), along with the protein intake per kg of body weight. The MD emphasised foods like cereals, extra virgin olive oil, vegetables, pulses, fruits, and blue fish, with limited intake of animal-origin foods, sugar, or alcohol. The caloric content of each participant’s personalised low-calorie MD was determined based on their individual dietary habits, ensuring a unique caloric deficit for each, which was essential, given that the caloric Δ is a pivotal variable in this study. The average macronutrient breakdown was 51% carbohydrates, 29% fats, 19% protein, and over 30 g of dietary fibre daily. Nutritional value tables were provided to assist participants with food choices. The caloric content of the diets was standardised for the purposes of this study, aiming to provide a uniform dietary framework for all the participants. This approach was chosen to maintain consistency across the study while still adhering to the principles of a low-calorie MD. The diets were designed to be nutritionally balanced and to facilitate a comparison of the effects of a standardised dietary intervention on BC and total energy expenditure across our participant cohort. To ensure the dietary approach was both practical and applicable in a real-world setting, the meal plans were created with consideration of typical dietary patterns and preferences within a Mediterranean dietary context. The caloric content of these diets was set within a range that is generally recommended for weight management and health promotion, reflecting current nutritional guidelines. Winfood 2.8 nutritional software (Medimatica Srl, Martinsicuro, Italy) was used for diet formulation. Regarding caloric intake, this study only reports dietary prescription data at T0. It is important to note that follow-up data (T1) on prescribed caloric intake are not available. This limitation should be considered in the interpretation of dietary intake changes and their impact on the study outcomes.

Furthermore, to provide a clearer understanding of the dietary intervention implemented, we have included an example of a typical day’s meal plan in the Mediterranean-style diet. This is provided in Appendix A. The meal plan outlines the portion sizes, ingredients, and total caloric content, exemplifying how a balanced, low-calorie diet was maintained while adhering to the principles of the MD.

### 2.4. Follow-Up

The prescribed diet and physical activity monitoring started at T0, with continuous monitoring by dieticians from T0 to T1 to ensure adherence to the diet and maintenance of physical activity levels. Participants completed a weekly diet diary from the start of the study, which included one weekend day. These diaries were collected and reviewed by dietitians to assess compliance. Bi-monthly consultations were scheduled for personalised nutritional advice and behavioural modification. A chat service was available during business hours for participants to consult dietitians. Weekly meetings with a dietitian focused on anthropometric measurements, BC, and nutrient intake evaluation. After two months, assessments were repeated to evaluate the effectiveness of the intervention.

### 2.5. Age Stratification and Participant Demographics

In this study, we recognised the importance of age as a determinant of metabolic rate and its potential impact on the effectiveness of the Mediterranean-style Diet (MD). Consequently, we stratified our participants into three distinct age groups to facilitate a more nuanced analysis of the diet’s impact across different life stages. These groups were defined as young adults (ages 18–29), middle-aged adults (ages 30–49), and older adults (ages 50–65). This stratification was based on standard demographic categorizations, which align with significant physiological and metabolic changes that typically occur within these age ranges. Additionally, we observed a gender disparity in our sample selection. This disparity arose due to the higher interest and willingness of female participants to engage in dietary intervention studies and the specific outreach methods employed, which yielded a greater response from female participants. The sample consisted of a higher percentage of women compared to men, reflecting these trends. While this may influence the generalizability of our findings, it underscores the real-world scenario of participant recruitment in nutrition research, where volunteer availability and response rates often vary by gender. We believe that considering these demographic variables—age and gender—is crucial for interpreting the study’s outcomes. Age-related metabolic differences can significantly influence the body’s response to dietary interventions, while gender-related factors may also play a role in dietary preferences and compliance. Therefore, this stratification not only enhances the validity of our findings but also underscores the need for personalised dietary recommendations that take into account these demographic factors.

### 2.6. Statistical Analyses

For statistical analyses, we utilised SPSS version 23.0 and Python 3.12, with seaborn, matplotlib, scikit-learn, Pandas, and SciPy for various functions including data visualization, machine learning, data manipulation, and statistical testing. The normality of data distribution was verified using the Kolmogorov–Smirnov test, followed by appropriate tests (independent sample t-tests or Mann–Whitney U tests) for baseline comparisons between genders. The Spearman correlation test analysed the relationship between Δ% FM-to-FFM ratio and Δ Kcal%. Heatmaps illustrated correlations among dietary factors and physical activities, considering only significant variables (*p* < 0.05). Data were presented as mean ± SD, with a significance threshold of *p* < 0.05. Multiple regression analyses, using a stepwise method, identified factors associated with Δ% FM-to-FFM ratio, Δ% FM, and Δ FFM. The analyses included a wide range of variables, measured at both T0 and T1, and adjusted for age, gender, and BMI T0. The large number of variables, some interrelated, necessitates cautious interpretation of the exploratory stepwise method results. The choice of the stepwise method in multiple regression analysis was guided by its ability to systematically select the most significant variables from a large pool, thereby reducing potential biases from multicollinearity. This method is particularly effective in studies like ours with numerous interrelated variables, as it helps to isolate the impact of each variable on the dependent outcomes.

## 3. Results

During the study period, 187 patients were evaluated at the medical centre. Of these, 87 individuals were excluded based on predefined criteria or their refusal to wear the armband. The final cohort for analysis comprised 100 patients who met the inclusion criteria and agreed to participate. Recruitment was halted upon reaching a sample size of 100 participants, as this provided a robust dataset for the retrospective analysis, ensuring sufficient statistical power to examine the study’s objectives. Baseline characteristics of the study participants are summarised in Table 1.

The study sample comprised 100 participants: 60 females and 40 males. The average age was 34.5 ± 10.6 years, with females averaging 34.7 ± 11.2 years and males averaging 34.3 ± 9.9 years. Anthropometric data showed an average height of 168.9 ± 8.9 cm, weight of 82.7 ± 20.0 kg, and BMI of 28.7 ± 5.4. The average FM was 25.9 ± 10.5 kg, representing 30.7 ± 7.8% of the total body mass. The mean FFM was 54 ± 12.9 kg. In addition to gender-specific analysis, this study further stratified participants based on age groups, aligning with metabolic differences across life stages. The distribution of participants was 30 in the young age group (18–29 years), 59 in the middle-aged group (30–49 years), and 11 in the older age group (50–65 years). This stratification allowed for more nuanced insights into how age influences the impact of diet and physical activity on the fat-to-lean mass ratio. The average caloric intake was 1792 Kcal, with males consuming 2143 Kcal and females consuming 1559 Kcal (Table 2).

Macronutrient distribution was approximately 20.1% protein, 28.9% fat, and 50.7% carbohydrates. The following gender differences were observed: males had more fat (29.7%), and females had more carbohydrates (51.2%). Average protein intake was 90 g (1.1 g/kg of body weight); males averaged 107 g, and females averaged 79 g. The average daily fibre intake was 34 g. Most calories were consumed during lunch (33.1%) and dinner (32%).

In our findings, as summarised in Table 3, we observed significant variations in Δ Kcal% and Δ% FM-to-FFM ratio across different demographic and physiological groups. Notably, the mean Δ Kcal% and Δ% FM-to-FFM ratio differed notably between genders, across various BMI categories, and among distinct age groups, indicating differential impacts of dietary intake and body composition changes. The results presented as Δ (T0 − T1) reflect the cumulative effects of the diet and physical activity regimen over the monitoring period, showcasing the changes in BC. Statistically significant changes in BC parameters across the entire sample (Table 4 and Appendix A). Specifically, the entire sample experienced a notable decrease in weight, BMI, FM, and TBW, alongside an increase in body proteins. There were no gender differences in the changes in FM-to-FFM ratio. Stratification by BMI shows a greater loss of FM in subjects with a higher BMI. Changes in body protein were non-significant within the groups. No significant differences were found between the age groups, except for the change in body protein. The 30–49-year-old and 50–65-year-old age groups had an increase in muscle mass in contrast to the younger age group in which there was no change.

In our study, we analysed Δ Kcal% through Spearman’s correlation to evaluate its relationship with the change in the fat mass to fat-free mass ratio (Δ% FM-to-FFM ratio). The correlation coefficient obtained was −0.04, signifying a very weak and statistically non-significant negative correlation. This result is visually represented in Figure 1. The low correlation coefficient suggests that the relationship between Δ Kcal% and the Δ% FM-to-FFM ratio was minimal and not as strong as hypothesised.

Figure 2 displays correlations among T0 dietary variables and the change in FM-to-FFM ratio from T0 to T1.

Figure 3 displays a Spearman correlation heatmap, illustrating the relationship between various physical activity parameters at baseline (T0) and the Δ% FM-to-FFM ratio. This heatmap provides insights into how initial physical activity levels may correlate with changes in body composition, specifically the fat mass to fat-free mass ratio, following dietary intervention. The visualization aids in identifying key baseline physical activity factors that could potentially influence the efficacy of the personalised low-calorie Mediterranean Diet in modifying body composition.

Figure 4 shows a weak inverse relationship (r = −0.21) between the Δ% FM-to-FFM ratio and average daily METs. Other physical activity parameters like step count and sleep duration show no significant correlations. Figure 4, reinforcing the observations in Figure 3, illustrates the relationship between changes in Δ% FM-to-FFM from T0 to T1 and the average METs recorded at T0. The weak negative correlation (r = −0.21, *p* = 0.04) suggests that higher exercise intensity may slightly contribute to a reduction in the ratio, indicating a trend towards improved BC.

Figure 5 depicts the correlation between physical activity duration (minutes, T0) and the Δ% FM-to-FFM ratio from T0 to T1. The analysis revealed a weak negative Spearman correlation (r = −0.20, *p* = 0.042).

Figure 6 displays a heatmap of Spearman correlation coefficients, highlighting the significant relationships between variables and the Δ% FM-to-FFM ratio across gender and BMI categories.

In the female subgroup, a pattern of significant negative correlations emerged, notably with g/kg protein intake (r = −0.27), exercise intensity as measured by METs (r = −0.29), physical activity duration (r = −0.27), and mid-morning calorie intake (R = −0.28). For individuals with a BMI ranging from 20–25, a significant positive relationship was found with fibre intake (r = 0.39). Conversely, those in the BMI 25–30 category demonstrated negative correlations with daily caloric intake (r = −0.51), protein consumption (r = −0.45), and intake of PUFA (r = −0.46) and MUFA, (r = −0.46), along with mid-morning calorie intake (r = −0.37). In the BMI category above 30, a negative correlation was observed with SFA (r = −0.37).

Multiple regression analysis, as shown in Table 5, indicates that age is a significant predictor of the Δ% FM-to-FFM ratio, with a regression coefficient of 0.13 (*p* = 0.036), highlighting the impact of age on BC changes. Specifically, the regression coefficient for age was 0.13, indicating a positive association with the Δ% FM-to-FFM ratio. The model’s adjusted R-squared value was 0.034, suggesting that, while age is a significant factor, it explains a small portion of the variance in the Δ% FM-to-FFM ratio.

The Δ%FM analysis (Table 6) identified “Energy Expenditure > 3 METs” as a significant factor inversely associated with Δ%FM. The coefficient of −0.0046 indicates that higher energy expenditure in activities over 3 METs is correlated with a decrease in the percent change in fat mass. The relatively low adjusted R-squared (0.045) value suggests that other factors not included in the model may also significantly influence changes in fat mass.

The stepwise multiple regression analysis for ΔFFM, encompassing variables such as age, gender, dietary intake, and physical activity, revealed no statistically significant predictors of change. This outcome persisted even after adjusting the selection criteria for the variables (initial *p*-value threshold <0.01 for entry and <0.05 for exit, adjusted to <0.05 for entry and <0.10 for exit). The final model included only the constant term, suggesting that either the relationships are not linear or other unmeasured variables might play a significant role.

In summary, the results demonstrate notable changes in BC parameters, with age emerging as a significant factor influencing these changes. While dietary and physical activity parameters show varying degrees of correlation with BC metrics, their predictive value appears to be moderated by other factors.

## 4. Discussion

Our study offers an understanding of the interplay between diet, physical activity, and changes in BC, specifically the Δ% FM-to-FFM ratio. Although existing literature underscores the significance of diet and physical activity in BC management [16,17,18], our findings challenge these conventional perspectives. Our analysis demonstrates the effectiveness of the prescribed interventions in altering BC, as evidenced by the changes in the Δ% FM-to-FFM ratio from T0 to T1. The data presented in Table 3 highlight crucial insights into how dietary changes and body composition alterations manifest differently across genders, BMI categories, and age groups. These variations underscore the importance of considering individual physiological and demographic factors when evaluating the effectiveness of dietary interventions and their impact on body composition. However, our analysis revealed that neither diet nor physical activity alone strongly predict changes in the Δ% FM-to-FFM ratio, indicating the influence of other factors beyond these traditional metrics (Figure 2 and Figure 3) [19,20]. These results are contrary to the widely held belief that stricter caloric restriction always results in improved BC [21]. This suggests that extreme dietary measures may not be required to achieve effective fat loss and muscle preservation. Such awareness could alleviate the psychological burden and potential metabolic consequences often associated with highly restrictive dietary regimes. A moderate caloric deficit may be just as beneficial for altering BC, adding another layer of flexibility and sustainability to weight management strategies (Figure 1, [22]). The absence of a definitive correlation, as shown in Figure 1, underscores the complexity of factors influencing the Δ% FM-to-FFM ratio beyond diet and physical activity alone. These could potentially include hormonal variations [23], metabolic rates [24], or even genetic predispositions [25], none of which were accounted for in this current study.

Subgroup analysis (Appendix A and Figure 6), especially in gender and BMI categories, revealed distinct trends that vary across gender and BMI categories, offering deeper insights into the differential impacts on the Δ% FM-to-FFM ratio. Our results revealed distinct patterns affecting the Δ% FM-to-FFM ratio, underscoring the necessity for personalised dietary and physical activity strategies. In the female subgroup, a cluster of variables exhibited negative correlations with the Δ% FM-to-FFM ratio. Specifically, increased protein intake per kg of body weight (R = −0.27), higher exercise intensity as measured by METs (R = −0.29), and a longer duration of physical activity (R = −0.27) were all associated with a more favourable Δ% FM-to-FFM ratio. Existing literature suggests that achieving a total daily protein intake of 1.4–2.0 g/kg/day may favour BC outcomes, irrespective of the protein source [26,27]. For individuals in the BMI range of 20–25 kg/m2, a counterintuitive finding was observed. Higher fibre intake was positively correlated with a less favourable Δ% FM-to-FFM ratio (R = 0.39). While fibre is generally considered beneficial for metabolic health, it might be that people in this BMI range consume fibre primarily from high-calorie sources or that there is an interaction effect with other nutrients or lifestyles, and its role in BC within this specific BMI range warrants further investigation [28]. In this subgroup, the reduced intake of dietary components such as daily caloric intake (r = −0.51), protein (R = −0.45), PUFA (R = −0.46), and MUFA (R = −0.46) were associated with a more favourable effect on the Δ% FM-to-FFM ratio. These findings could provide the basis for targeted nutritional interventions aimed at individuals within this BMI range [29]. Interestingly, in obese individuals, a reduction in SFA was correlated with a more favourable Δ% FM-to-FFM ratio (R = −0.37). This confirms existing concerns about the negative impacts of saturated fats on metabolic health and BC, particularly in obese individuals [30,31].

The finding that age is significantly associated with the percentage change in the ratio of fat mass to lean mass (Δ% FM-to-FFM ratio) underscores the complexity of BC dynamics across different life stages [32]. The positive association between age and the Δ% FM-to-FFM ratio, particularly pronounced in the older adult cohort, suggests age-specific metabolic and compositional changes that warrant further investigation [33]. Notably, the significant improvement in body muscle mass among the older adult cohorts is particularly striking, suggesting that these age groups might be more responsive to muscle mass or protein balance modifications induced by dietary changes [34]. This age-specific response highlights the necessity of tailoring research and interventions to address the distinct BC needs of older adults [35]. Furthermore, our study reveals an inverse relationship between energy expenditure in activities over three METs and the Δ% FM-to-FFM ratio, indicating that higher-intensity physical activity could be a key factor in reducing fat mass percentage [34]. This observation aligns with existing literature, which posits that increased—especially more intense—physical activity is crucial for optimal BC and FM management [35]. Similarly, the duration of physical activity, though intuitively impactful, showed only a marginal effect on BC changes, suggesting that simply increasing exercise time might not be as effective as previously thought. However, the low adjusted R-squared value in our analysis hints at the presence of other unmeasured variables that might significantly influence these ratios, pointing to the multifaceted nature of BC changes and the need for comprehensive research approaches.

Our study has limitations, including a limited sample size, retrospective nature, and the absence of several potentially influencing variables like hormonal status, general health condition, and psychological factors. Another limitation is the absence of a formal power analysis to determine the sample size. The sample size was primarily determined based on available data and logistical considerations. We acknowledge that power analysis is crucial for ensuring an adequately powered study, which is essential for detecting true effects and for the statistical significance of the results. While our findings are insightful, the limited sample size and retrospective nature of the study may restrict the generalizability of the results, underscoring the need for larger, more diverse cohorts in future research. A control group could have provided a comparative baseline for more definitive conclusions. The two-month follow-up period, while brief, was deemed sufficient for initial observation of the MD and physical activity impacts on the FM-to-FFM ratio. Comparable studies [36,37] have also observed significant changes within similar timeframes. Nevertheless, we acknowledge that longer follow-up periods could provide more extensive data.

## 5. Conclusions

Our study challenges the conventional paradigm that places diet and physical activity as exclusive determinants of changes in the Δ% FM-to-FFM ratio. Although we did not identify unique and unequivocal predictive factors, the intricate relationships and variations observed between subgroups lead us to re-evaluate the simplistic “one-size-fits-all” approach to BC management. Our results contribute significantly to a refined understanding of the nuanced interaction between dietary patterns and physical activity levels in shaping BC. Rather than viewing the absence of a single dominant factor as inconclusive, our study highlights the multifaceted nature of BC dynamics, emphasising the need for personalised, multifactorial approaches in nutrition and exercise science. This inherent complexity emphasises the imperative for personalised approaches in the development of dietary and exercise prescriptions, with careful consideration to the divergent impacts observed among different subpopulations. Furthermore, our study represents an initial exploration, providing the basis for more comprehensive investigations in the future. Subsequent research should further investigate the influence of factors such as hormonal variations, metabolic rates, and psychological determinants on changes in BC. These efforts promise to acquire a more complete understanding of the intricate interactions, influencing the development of customised strategies for optimal BC management.

## Figures and Tables

**Figure 1 nutrients-16-00019-f001:**
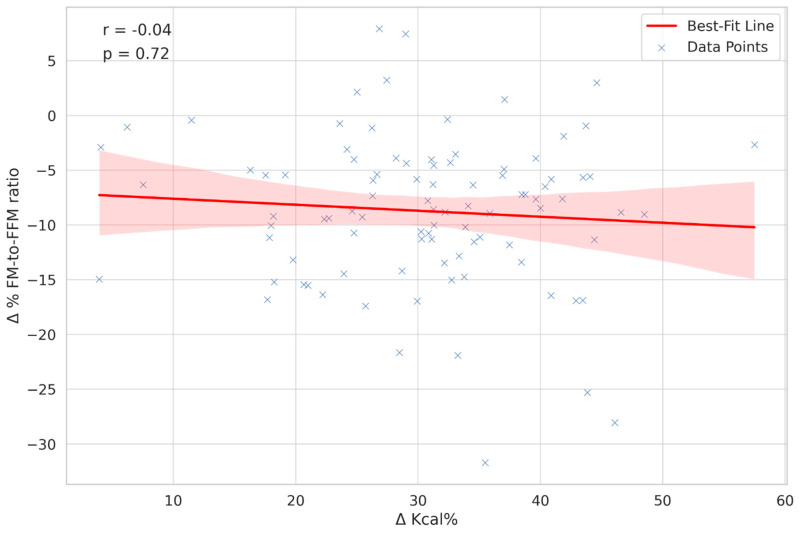
Correlation analysis between caloric intake change and body composition shift. Figure 1 illustrates the correlation between Δ Kcal% and Δ% FM-to-FFM ratio. A scatter plot is presented, with a best-fit line to demonstrate the correlation trend among 100 subjects. Spearman’s correlation coefficient of −0.04 suggests a very weak negative correlation, underlining the complexity of the relationship. Abbreviations are elucidated as follows: Δ Kcal% = Percent change in caloric intake; Δ% FM-to-FFM ratio = Percent change in fat mass to fat-free mass ratio. The Δ Kcal% is depicted as an absolute value to illustrate its correlation with the Δ FM-to-FFM ratio%. This approach allows for a clear visualization of the relationship between changes in caloric intake and alterations in body composition. The presentation of Δ Kcal% as an absolute value highlights the varying degrees of caloric intake change, either increase or decrease, among participants.

**Figure 2 nutrients-16-00019-f002:**
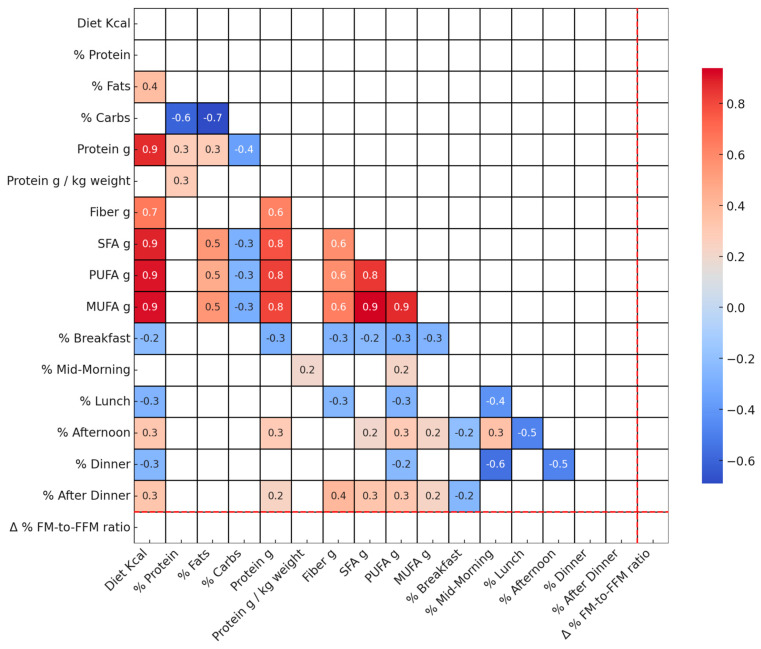
Spearman correlation heatmap of dietary variables at baseline and Δ% FM-to-FFM ratio. This figure displays Spearman correlation coefficients among dietary variables measured at T0 and the change in the FM-to-FFM ratio between T0 and T1. Correlations significant at a *p*-value less than 0.05 are shown. The following variables are included: SFA g: Saturated Fats, PUFA g: Polyunsaturated Fats, MUFA g: Monounsaturated Fats, FM: Fat Mass, FFM: Fat-Free Mass.

**Figure 3 nutrients-16-00019-f003:**
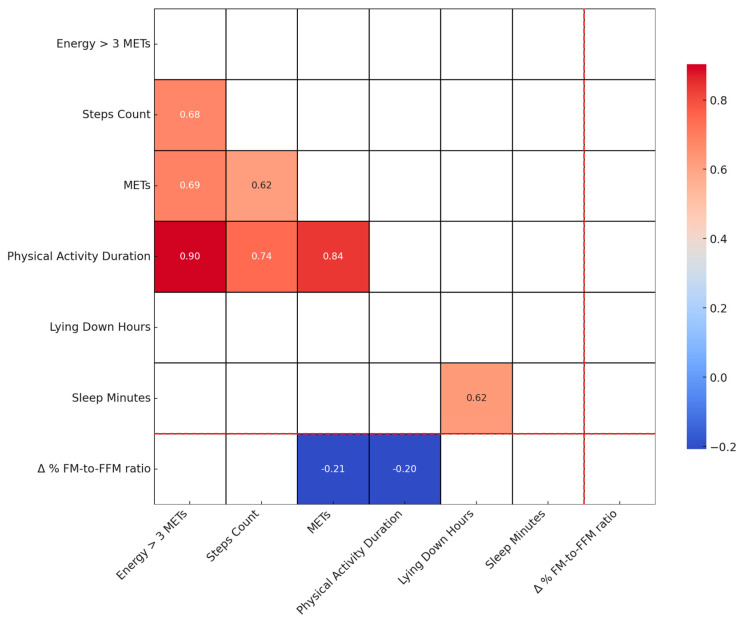
Spearman correlation heatmap of physical activity parameters at baseline and Δ% FM-to-FFM ratio. This Figure shows Spearman correlation coefficients between physical activity parameters measured at baseline (T0) and the percentage change in FM-to-FFM ratio from T0 to T1. Only correlations with a *p*-value less than 0.05 are displayed.

**Figure 4 nutrients-16-00019-f004:**
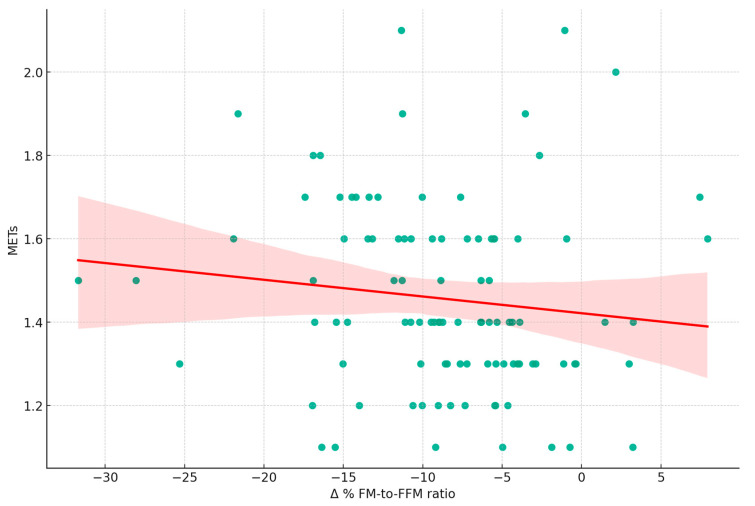
Relationship between changes in fat mass to fat-free mass ratio and exercise intensity. This scatter plot illustrates the relationship between the percentage change in Fat Mass to Fat-Free Mass Ratio (Δ% FM-to-FFM ratio) from T0 to T1 and average Metabolic Equivalent of Task (METs) measured at T0 (r = −0.20, *p* = 0.042). The following variables are included: FM: Fat Mass; FFM: Fat-Free Mass; METs: Metabolic Equivalent of Task; Δ% FM-to-FFM ratio.

**Figure 5 nutrients-16-00019-f005:**
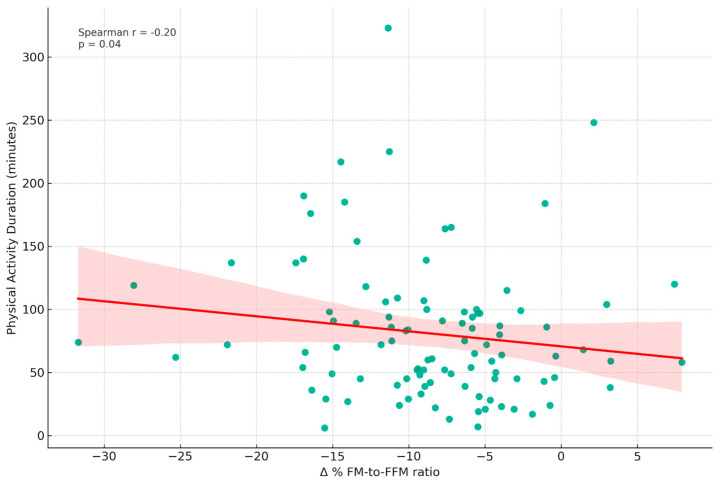
Correlation between physical activity duration and changes in fat mass to fat-free mass ratio. This scatter plot displays the relationship between Physical Activity Duration (in minutes) measured at T0 and the Δ% FM-to-FFM ratio from T0 to T1. A weak negative Spearman correlation (r = −0.20, *p* = 0.042) was observed. The following variables were included: FM: Fat Mass; FFM: Fat-Free Mass.

**Figure 6 nutrients-16-00019-f006:**
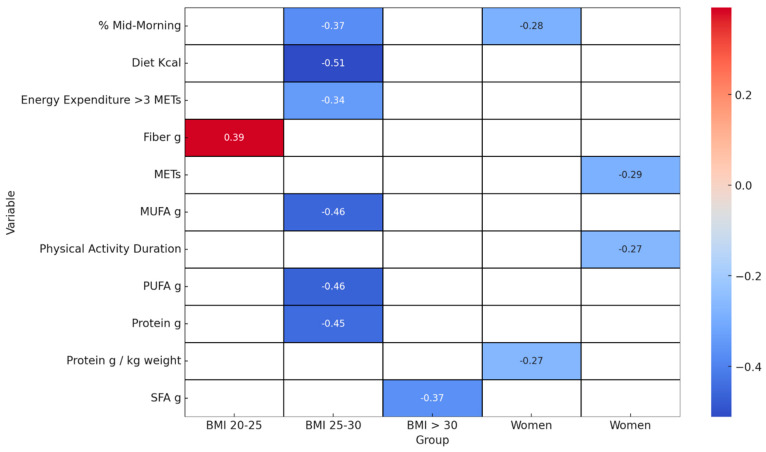
Significant Spearman correlations of the Δ% FM-to-FF ratio: comparison of dietary and sport variables and BMI categories and gender. This heatmap of Spearman correlation coefficients focuses on the significant relationships between various variables and the change in the percentage of fat mass to fat-free mass (Δ% FM-to-FFM ratio) across different gender and BMI categories. Only statistically significant values are included, and these are as follows: SFA g: Saturated Fats in grams, PUFA g: Polyunsaturated Fats in grams, MUFA g: Monounsaturated Fats in grams.

**Table 1 nutrients-16-00019-t001:** Summary of demographic, anthropometric, activity, and biochemical data at T0.

Variable	Total (100)	Females (60)	Males (40)	Age 18–29 (30)	Age 30–49 (59)	Age 50–65 (11)
Age	34.5 ± 10.6	34.7 ± 11.2	34.3 ± 9.9	22.9 ± 3.9	37.1 ± 5.4	53.6 ± 2.9
Smokers	15	10	5	5	8	2
Anthropometric Data						
Height (cm)	168.9 ± 8.9	163.9 ± 5.8	176.6 ± 7.0	170.1 ± 8.5	169.6 ± 8.8	162.1 ± 8.4
Weight (kg)	82.7 ± 20.0	73.1 ± 14.5	97.0 ± 18.6	77.6 ± 13.4	84.5 ± 29	86.7 ± 20.1
BMI (Kg/m^2^)	28.7 ± 5.4	27.2 ± 5.1	31.0 ± 5.2	26.1 ± 2.4	29 ± 5.7	32.8 ± 6.3
FM (Kg)	25.9 ± 10.5	25.4 ± 10.4	26.5 ± 10.8	21.9 ± 7.7	26.3 ± 10.4	33.9 ± 13.1
FM (%)	30.7 ± 7.8	33.6 ± 6.9	26.3 ± 7.1	28.1 ± 8.2	30.6 ± 6.9	38.1 ± 7.2
FFM (Kg)	54 ± 12.9	45.3 ± 5.3	67.1 ± 9.5	53.0 ± 10.9	55.3 ± 14.4	50.1 ± 9.3
FM-to-FFM ratio	0.49 ± 0.18	0.55± 0.18	0.39 ± 0.14	0.43 ± 0.17	0.48 ± 0.16	0.67 ± 0.22
Body Water (Kg)	40.4 ± 9.7	33.8 ± 4.4	50.2 ± 6.8	39.6 ± 7.6	41.2 ± 10.9	37.9 ± 8
Body Proteins	13.4 ± 3.6	11.1 ± 1.4	16.9 ± 3.2	13.2 ± 3.4	13.9 ± 3.9	11.4 ± 2.3
Metabolic and Activity Data						
BEE (Kcal)	1773 ± 458	1437 ± 164	2173 ± 361	1731 ± 442	1806 ± 493	1692 ± 358
TEE	2624 ± 566	2263 ± 270	3166 ± 449	2643 ± 489	2656 ± 611	2405 ± 509
LAF	1.5 ± 0.2	1.4 ± 0.2	1.5 ± 0.2	1.5 ± 0.2	1.4 ± 0.2	1.3 ± 0.2
Energy > 3 METS (kcal)	432 ± 292	315 ± 205	607 ± 318	502 ± 281	430 ± 312	280 ± 204
Weekly Sport Days (days/week)	2.6 ± 1.6	2.6 ± 1.5	2.7 ± 1.8	2.9 ± 1.4	2.5 ± 1.7	2.3 ± 1.9
Daily Steps (n)	9574 ± 3875	8828 ± 3221	10,693 ± 4503	10,351 ± 4735	9499 ± 3705	8800 ± 2763
Average Sports Duration (h:mm)	1.21 ± 0.55	1.08 ± 51.0	1.40 ± 0.56	1.36 ± 0.52	1.20 ± 0.59	0.51 ± 0.31
Lying Down (h:mm)	7.47 ± 1.12	7.55 ± 1.12	7.37 ± 1.13	7.58 ± 1.17	7.39 ± 1.12	8.18 ± 1.07
Sleep (h:mm)	6.21 ± 0.55	6.33 ± 0.52	6.01 ± 0.55	6.23 ± 1.08	6.16 ± 0.50	6.28 ± 0.56
Biochemical Data						
Glycemia	86.8 ± 12.2	83.8 ± 9.8	91.2 ± 14.1	82.9 ± 10.9	87.4 ± 12.3	92.8 ± 12.6
TC	194.3 ± 49.4	202.2 ± 49.8	183.5 ± 47.6	180.9 ± 62.6	192.1 ± 33.1	225.4 ± 54.6
HDL	56.5 ± 16.0	61.7 ± 15.2	49.7 ± 14.9	60.0 ± 14.6	54.0 ± 16.2	57.0 ± 18.2
TG	108.2 ± 65.3	89.8 ± 39.9	132.3 ± 83.1	93.9 ± 46.3	104.5 ± 51.4	142.0 ± 89.3
SBP	123.3 ± 14.1	119.5 ± 12.4	129.1 ± 14.8	121.6 ± 14.7	123.1 ± 14.0	128.7 ± 13.4
DBP	77.0 ± 9.1	74.9 ± 9.2	80.1 ± 8.2	74.7 ± 8.1	77.1 ± 9.6	81.9 ± 7.1

Table 1 summarises demographic, anthropometric, activity, and biochemical data, providing detailed insights into the baseline characteristics of the study participants. The table includes the total number of participants, broken down into females, males, and age. Data are presented as the mean ± standard deviation. The following body composition parameters were measured at both the initial (T0) and final (T1) time points of the study: Weight, BMI, Fat Mass (both in kilograms and as a percentage of total body weight), Fat-Free Mass, the Fat Mass-to-Fat-Free Mass ratio, Total Body Water (TBW), and Body Protein. Other parameters, including Basal Energy Expenditure, Total Energy Expenditure, Light Activity Frequency, energy expenditure above 3 METs, weekly sport participation days, daily step count, average sport duration, lying down time, sleep duration, Glycemia, Total Cholesterol (TC), High-Density Lipoprotein (HDL), Triglycerides (TG), Systolic Blood Pressure (SBP), and Diastolic Blood Pressure (DBP) were assessed only at baseline (T0). The “Body Proteins” in Table 1 refer to the total protein content in the body anthropometrics. Sample size for analysis: 100 subjects in a retrospective analysis. Basal Energy Expenditure (BEE), Total Energy Expenditure (TEE), Level of Activity Factor (LAF), Energy Dispensed at Activity > 3 METS (Energy > 3 METS), Average Sports Duration (ASD). Biochemical Data includes blood markers like Total Cholesterol (TC), High-Density Lipoprotein (HDL), Triglycerides (TG), SBP, and DBP.

**Table 2 nutrients-16-00019-t002:** Nutritional parameters at T0.

Variable	Total (100)	Females (60)	Males (40)	*p*-Value
Energy kcal	1792.24 ± 401.53	2142.85 ± 360.98	1558.50 ± 213.31	<0.001
% Protein	20.12 ± 2.30	19.97 ± 2.90	20.22 ± 1.82	0.44
% Fats	28.90 ± 2.49	29.71 ± 2.80	28.37 ± 2.13	0.01
% Carbs	50.74 ± 2.98	50.02 ± 3.51	51.22 ± 2.50	0.05
Protein g	90.20 ± 20.49	106.53 ± 19.50	79.21 ± 12.17	<0.001
Protein g/kg weight	1.11 ± 0.22	1.13 ± 0.27	1.10 ± 0.19	0.45
Fiber g	33.98 ± 7.05	37.87 ± 6.57	31.36 ± 6.12	<0.001
SFA g	12.08 ± 4.32	15.42 ± 4.41	9.83 ± 2.38	<0.001
PUFA g	7.07 ± 4.08	8.67 ± 4.70	5.99 ± 3.21	0.00
MUFA g	29.50 ± 8.00	36.21 ± 6.93	24.99 ± 4.94	<0.001
% Breakfast	16.70 ± 3.42	16.29 ± 3.45	16.97 ± 3.40	0.25
% Mid-Morning	7.52 ± 6.93	7.84 ± 6.90	7.30 ± 7.00	0.50
% Lunch	33.09 ± 6.80	32.98 ± 6.71	33.16 ± 6.92	0.63
% Afternoon	9.67 ± 6.61	10.28 ± 6.74	9.27 ± 6.54	0.33
% Dinner	31.97 ± 7.21	31.19 ± 7.80	32.48 ± 6.82	0.29
% After Dinner	1.04 ± 2.24	1.39 ± 2.29	0.81 ± 2.19	0.16

The table presents the mean and standard deviation (SD) of nutritional parameters, including caloric intake; the percentage of protein; fat; carbohydrates; protein intake in grams; protein intake per kilogram of body weight; fibre intake; and saturated, polyunsaturated, and monounsaturated fats. Sample size for analysis: 100 subjects in a retrospective analysis. Baseline Dietary Parameters (T0): This table details the dietary prescriptions given to participants at the beginning of the study, including caloric content and the breakdown of macronutrients and meal-time distribution. Values are broken down by gender: total, male, and female. Also included are values of *p* obtained from Student’s *t*-tests for comparing males and females for each nutritional parameter. Values of *p* less than 0.05 indicate statistically significant differences between the two groups. SFA g: Saturated Fats, PUFA g: Polyunsaturated Fats, MUFA g: Monounsaturated Fats.

**Table 3 nutrients-16-00019-t003:** Descriptive statistics of Δ Kcal% and Δ% FM-to-FFM ratio across different subgroups.

	Δ Kcal%	Δ% FM-to-FFM Ratio
Total	31.0 ± 10.2	−8.6 ± 6.8
Female	30.5 ± 10.2	−7.9 ± 6.7
Male	31.7 ± 10,2	−9.7 ± 6.8
BMI 18.5–24.9	29.0 ± 9.1	−10.3 ± 7.6
BMI 25–29.9	31.0 ± 11.1	−7.5 ± 5.3
BMI 30–34.9	32.8 ± 9.4	−8.7 ± 8.1
Age 18–29	30.2 ± 10.3	−8.9 ± 5.4
Age 30–49	31.3 ± 10.0	−9.6 ± 6.9
Age 50–65	32.1 ± 11.1	−2.2 ± 6.2

This table presents the mean ± standard deviation for Δ Kcal% (change in caloric intake percentage) and Δ% FM-to-FFM ratio (change in fat mass to fat-free mass ratio percentage) across the total sample, gender-specific subgroups, different BMI categories, and age groups. The data provide insights into the variations in dietary intake and body composition changes among diverse demographic and physiological segments of the study population. Data are presented as average ± SD.

**Table 4 nutrients-16-00019-t004:** Changes in various body composition parameters (T1 − T0).

						BMI				Age Groups			
Variable	Whole Sample	*p*-Value	Females	Males	*p*-Value (M vs. F)	18.5–24.9	25–29.9	30–34.9	*p*-Value (BMI)	18–29	30–49	50–65	*p*-Value (Age Groups)
WEIGHT (kg)	−2.9 ± 1.8	<0.0001	−2.5 ± 1.8	−3.5 ± 1.7	0.003	−2.1 ± 1.5	−2.5 ± 1.2	−3.6 ± 1.6	<0.001	−2.9 ± 1.3	−3.0 ± 2.0	−2.6 ± 1.9	0.895
BMI (Kg/m^2^)	−1.0 ± 0.6	<0.0001	−0.9 ± 0.6	−1.1 ± 0.5	0.076	−0.8 ± 0.6	−0.9 ± 0.4	−1.2 ± 0.6	<0.001	−1 ± 0.4	−1.0 ± 0.7	−1.0 ± 0.7	0.747
FM (kg)	−2.4 ± 1.7	<0.0001	−2.1 ± 1.4	−2.8 ± 2.0	0.022	−1.7 ± 1.1	−2.0 ± 0.9	−2.9 ± 2.0	<0.001	−2.1 ± 1.0	−2.6 ± 1.9	−1.6 ± 1.7	0.440
FM (%)	−1.8 ± 1.5	<0.0001	−1.7 ± 1.4	−1.9 ± 1.6	0.494	−2.0 ± 1.5	−1.6 ± 1.0	−1.9 ± 2.0	<0.001	−1.8 ± 1.1	−2.1 ± 1.6	−0.7 ± 1.3 *	0.110
FFM (kg)	−0.5 ± 1.4	0.0012	−0.4 ± 1.2	−0.7 ± 1.7	0.320	−0.4 ± 1.0 *	−0.5 ± 1.1	−0.6 ± 1.8 *	<0.001	−0.7 ± 1.0	−0.3 ± 1.6 *	−1.2 ± 1.6	0.091
FM-to-FFM ratio	−0.04 ± 0.03	<0.0001	−0.04 ± 0.03	−0.04 ± 0.04	0.865	−0.04 ± 0.03	−0.04 ± 0.03	−0.05 ± 0.05	<0.001	−0.04 ± 0.02	−0.05 ± 0.03	−0.02 ± 0.04 *	0.130
TBW (kg)	−0.6 ± 1.0	<0.0001	−0.4 ± 1.0	−0.8 ± 0.9	0.014	−0.4 ± 0.8	−0.4 ± 0.8	−0.9 ± 1.1	<0.001	−0.5 ± 1.0	−0.5 ± 1.0	−1.0 ± 1.0	0.161
Body Protein (Kg)	0.3 ± 1.0	0.0049	0.4 ± 1.0	0.2 ± 1.0 *	0.395	0.1 ± 0.7 *	0.2 ± 0.8 *	0.3 ± 1.1 *	0.007	−0.01 ± 0.6 *	0.4 ± 1.1	0.6 ± 1.0 *	0.036

This table presents the changes in various body composition parameters between two time points (T0 and T1). Data are presented as mean differences ± standard deviation. *p*-values are provided for changes within the whole sample, between males and females, and among different age groups. Independent sample t-tests were used to determine the statistical significance for the *p*-values presented. * Indicates that the differences for the variables between T0 and T1 are not significant (see Appendix A).

**Table 5 nutrients-16-00019-t005:** Multiple regression analysis showing the relationship between various factors and the Δ% FM-to-FFM ratio. Only statistically significant parameters are shown.

	Beta	*p*-Value	95% CI
Constant	−13.23	<0.001	(−17.73, −8.73)
Age	0.13	0.036	(0.01, 0.26)

**Table 6 nutrients-16-00019-t006:** Multiple regression analysis showing the relationship between various factors and the Δ%FM. Only statistically significant parameters are shown.

	Beta	*p*-Value	95% CI
Constant	−7.48	<0.001	(−9.50, −5.46)
Energy Expenditure > 3 METs	−0.0046	0.019	(−0.009, −0.001)

## Data Availability

The data that support the findings of this study are available from the corresponding author, M.L., upon reasonable request.

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
