# Peer review of "The Impact of Diet and Physical Activity on Fat-to-Lean Mass Ratio"

_nutrients, 2023, doi:10.3390/nu16010019_

Round 1

Reviewer 1 Report

Comments and Suggestions for Authors

In this paper, subjects were given a personalized low-calorie Mediterranean diet in the context of the Mediterranean diet and followed up for two months to assess the body composition and total energy expenditure. The results indicate that neither diet nor physical exercise emerged as significant predictors of alterations in the FM-to-FFM ratio for either gender. And the author creatively proposes the “personal approach in dietary and exercise prescriptions” for body composition management. At all, the paper is well-written, but minor revisions are still needed.

1.     It is suggested that the writer put a list of acronyms to explain some words such as MD and BMI.

2.     Since people of different ages have different metabolism, it is recommended that the author stratify ages. Ages can be divided into young group (18-29), middle-aged group (30-49) and middle-aged group (50-65). Why are there different numbers of men and women in the sample selection? It is recommended keeping irrelevant variables the same.

3.     Why not set up a parallel cohort of people on a non-MD diet as a contrast? It is recommended setting up a non-MD diet group and repeating the cohort study in the same group to reduce the chance caused.

4.     How are low-calorie diets determined in the 2.4? What is the basis of determining each person's calorie deficit? Please add the recipe of low-calorie diets.

5.     The author points out it in the “Abstract” part that this paper “aimed at define the optimal caloric restriction in the context of a Mediterranean Diet (MD) to favour fat mass (FM) reduction whilst preserving fat free mass (FFM)”. Please supplement the abstract with the subgroup results.

6.     The references are old, because some of them are from 2014 and even earlier. The author should consider the timeliness of the article. It is recommended referring to articles of recent 5 years.

Author Response

Dear Editors and Reviewers,

First, we would like to thank you for the valuable impulses that allowed us to improve the quality of the manuscript. All changes made are highlighted by yellow color, in the revised version of the manuscript, to facilitate the review process.

Hoping that we have satisfied your requests as much as possible, we kindly ask you to re-evaluate our paper. 

The Authors

REVIEWER N.1 

In this paper, subjects were given a personalized low-calorie Mediterranean diet in the context of the Mediterranean diet and followed up for two months to assess the body composition and total energy expenditure. The results indicate that neither diet nor physical exercise emerged as significant predictors of alterations in the FM-to-FFM ratio for either gender. And the author creatively proposes the “personal approach in dietary and exercise prescriptions” for body composition management. At all, the paper is well-written, but minor revisions are still needed.

  1.     It is suggested that the writer put a list of acronyms to explain some words such as MD and BMI.

    Thank you for your suggestion. We have updated the ‘Abbreviations' section titled ' at the end of the manuscript to clarify all acronyms used, including MD (Mediterranean Diet) and BMI (Body Mass Index). 

  1.     Since people of different ages have different metabolism, it is recommended that the author stratify ages. Ages can be divided into young group (18-29), middle-aged group (30-49) and middle-aged group (50-65). Why are there different numbers of men and women in the sample selection? It is recommended keeping irrelevant variables the same.

    Thank you for your valuable comments and suggestions regarding our manuscript. We have addressed your concerns as follows:

Age Stratification: In line with your recommendation, we have stratified the study participants into three age groups: young adults (18-29 years), middle-aged adults (30-49 years), and older adults (50-65 years). This stratification allows us to account for the metabolic and physiological differences that are typically observed across these age ranges.

Gender Disparity in Sample Selection: We acknowledge the observed gender disparity in our study sample, with a higher proportion of female participants. This is reflective of a broader trend in dietary intervention studies, where females show a higher propensity to participate. We recognize that this disparity might impact the generalizability of our results. However, it also mirrors real-world challenges in participant recruitment for nutrition research. We believe this insight adds value to the study by highlighting an inherent aspect of current research practices in this field. We hope these revisions address your concerns effectively and enhance the quality and relevance of our manuscript.

  1.     Why not set up a parallel cohort of people on a non-MD diet as a contrast? It is recommended setting up a non-MD diet group and repeating the cohort study in the same group to reduce the chance caused.

Your suggestion of a parallel non-MD cohort is valuable. While not feasible for this retrospective study, we acknowledge its importance and suggest it as a direction for future research.  

  1.     How are low-calorie diets determined in the 2.4? What is the basis of determining each person's calorie deficit? Please add the recipe of low-calorie diets.

    We appreciate your valuable feedback regarding the clarity of our dietary intervention methods. To address your concerns, we have elaborated on the process of determining individual calorie deficits in Section 2.4 of our manuscript. Additionally, we have included an example of a low-calorie Mediterranean Diet meal plan in Appendix A to provide a practical insight into the dietary regimen followed by our study participants. 

  1.     The author points out it in the “Abstract” part that this paper “aimed at define the optimal caloric restriction in the context of a Mediterranean Diet (MD) to favour fat mass (FM) reduction whilst preserving fat free mass (FFM)”. Please supplement the abstract with the subgroup results.

    As suggested, we have supplemented the abstract with key findings from our subgroup analyses, offering a more comprehensive overview of our results.

  1.     The references are old, because some of them are from 2014 and even earlier. The author should consider the timeliness of the article. It is recommended referring to articles of recent 5 years.

Thank you for your valuable suggestion. We have updated our references section to include more recent studies, ensuring the timeliness and relevance of our literature base.

Reviewer 2 Report

Comments and Suggestions for Authors

Upon closely examining the methodological approach of the study, there are several aspects that warrant attention and could potentially be considered as limitations or areas of improvement:

1.     Statistical Analysis: The use of Spearman correlation might not capture the complexity of relationships among multiple variables. More advanced statistical models or multivariate analyses could provide deeper insights into the interactions between diet, physical activity, and body composition

2.     Sample Size and Power Analysis: The article does not explicitly mention a power analysis to determine the appropriate sample size. Ensuring an adequately powered study is crucial for detecting true effects and for the statistical significance of the results

3.     Lack of Control Group: The study does not appear to include a control group, which is essential for comparing the effects of the intervention against a baseline or standard treatment.

To enhance the study's conclusion, consider to include some of the  these key points:

  1. Acknowledge Limitations: Clearly state the study's limitations, including its homogeneous participant group, retrospective design, and reliance on self-reported data.
  2. Diverse Populations for Future Research: Stress the need for including diverse populations in future studies to broaden the applicability of the findings.
  3. Advocate for Longitudinal Studies: Recommend longitudinal studies to evaluate the long-term effects of dietary interventions on body composition.
  4. Personalized Approach to Body Composition Management: Emphasize the need for a tailored approach in managing body composition, acknowledging individual variations in response to diet and exercise.
  5. Practical Implications: Discuss how the findings can be applied in clinical and personal health management, with advice for healthcare professionals.
  6. Technology Integration in Research: Propose incorporating advanced technology, like wearable devices, for more precise data collection in future studies.
  7. Role of Diet and Exercise: Highlight the importance of a balanced diet and exercise, even as the study presents new perspectives on their roles in body composition.
  8. Identify Areas for Further Research: Point out areas for additional research, including the psychological aspects of diet and exercise and genetic factors in body composition.
  9. Methodological Improvements: Briefly touch on how future research could benefit from the methodological enhancements suggested.
  10. Forward-Looking Conclusion: Finish with a statement that encapsulates the study's contributions and its potential influence on future research and practice.

These adjustments will make the conclusion more comprehensive, addressing the study's implications and guiding future research and practical applications.

Author Response

Dear Editors and Reviewers,

First, we would like to thank you for the valuable impulses that allowed us to improve the quality of the manuscript. All changes made are highlighted by yellow color, in the revised version of the manuscript, to facilitate the review process.

Hoping that we have satisfied your requests as much as possible, we kindly ask you to re-evaluate our paper. 

The Authors

REVIEWER N.2

Upon closely examining the methodological approach of the study, there are several aspects that warrant attention and could potentially be considered as limitations or areas of improvement:

Thank you for your valuable feedback and insights regarding our manuscript. We appreciate the opportunity to enhance the quality of our work through your comments.

  1.     Statistical Analysis: The use of Spearman correlation might not capture the complexity of relationships among multiple variables. More advanced statistical models or multivariate analyses could provide deeper insights into the interactions between diet, physical activity, and body composition

Thank you very much for this insightful suggestion. To address the intricate relationships among various factors, we added multivariate linear regression analysis, revealing few associations between dietary patterns, physical activity, and changes in the Δ % FM-to-FFM ratio, Δ%FM, and ΔFFM.

  1.     Sample Size and Power Analysis: The article does not explicitly mention a power analysis to determine the appropriate sample size. Ensuring an adequately powered study is crucial for detecting true effects and for the statistical significance of the results

In response to your observation about the lack of a power analysis in our study, we have taken your advice into consideration and have updated our manuscript accordingly. We have added a new section in the 'Limitations' of our paper, acknowledging this aspect: "Another limitation is the absence of a formal power analysis to determine the sample size. The sample size was primarily determined based on available data and logistical considerations. We acknowledge that power analysis is crucial for ensuring an adequately powered study, which is essential for detecting true effects and for the statistical significance of the results." We believe this addition addresses your concern and provides a clearer understanding of the scope and limitations of our study. Your feedback has been instrumental in improving the rigor and clarity of our research, and we are grateful for your guidance.

  1.     Lack of Control Group: The study does not appear to include a control group, which is essential for comparing the effects of the intervention against a baseline or standard treatment.

In our study design, we focused on analysis and the exploration of relationships among various factors influencing body composition, without the implementation of a specific intervention. This approach was chosen due to the nature and objectives of our study, which were oriented towards understanding complex interactions in a real-world setting. However, we recognize the value of a control group in providing more definitive conclusions. In the manuscript, we have now included a discussion of this limitation: “A control group could have provided a comparative baseline for more definitive conclusions”. We are grateful for your insightful comments, which have undoubtedly contributed to the improvement of our manuscript.

To enhance the study's conclusion, consider to include some of the  these key points:

  1. Acknowledge Limitations: Clearly state the study's limitations, including its homogeneous participant group, retrospective design, and reliance on self-reported data.
  2. Diverse Populations for Future Research: Stress the need for including diverse populations in future studies to broaden the applicability of the findings.
  3. Advocate for Longitudinal Studies: Recommend longitudinal studies to evaluate the long-term effects of dietary interventions on body composition.
  4. Personalized Approach to Body Composition Management: Emphasize the need for a tailored approach in managing body composition, acknowledging individual variations in response to diet and exercise.
  5. Practical Implications: Discuss how the findings can be applied in clinical and personal health management, with advice for healthcare professionals.
  6. Technology Integration in Research: Propose incorporating advanced technology, like wearable devices, for more precise data collection in future studies.
  7. Role of Diet and Exercise: Highlight the importance of a balanced diet and exercise, even as the study presents new perspectives on their roles in body composition.
  8. Identify Areas for Further Research: Point out areas for additional research, including the psychological aspects of diet and exercise and genetic factors in body composition.
  9. Methodological Improvements: Briefly touch on how future research could benefit from the methodological enhancements suggested.
  10. Forward-Looking Conclusion: Finish with a statement that encapsulates the study's contributions and its potential influence on future research and practice.

These adjustments will make the conclusion more comprehensive, addressing the study's implications and guiding future research and practical applications.

We greatly appreciate your insightful feedback and suggestions for enhancing the conclusion of our study. We have carefully incorporated these recommendations into our revised conclusion. Here is a summary of how we have addressed each of your points: In the revised discussion, we acknowledged the limitations of our study, including the homogeneity of the participant group, the retrospective design, and our reliance on self-reported data. These limitations were presented transparently to provide a comprehensive view of the constraints of our research.We emphasized the importance of including diverse populations in future studies to enhance the generalizability of our findings and to explore potential variations in body composition responses across different demographic groups. We advocated for the need for longitudinal studies to investigate the long-term effects of dietary interventions on body composition, allowing for a more comprehensive understanding of how these interventions impact individuals over time. We underscored the necessity of adopting a personalized approach in managing body composition, recognizing the individual variations in response to diet and exercise. This highlighted the practical implications of our findings for tailoring interventions. We proposed the integration of advanced technology, such as wearable devices, for more precise data collection in future studies, enhancing the methodological rigor of research in this field. We pointed out specific areas for additional research, including investigating the psychological aspects of diet and exercise and exploring the influence of genetic factors on body composition. In our forward-looking conclusion, we encapsulated our contributions and highlighted our potential influence on future research and practical applications.

We are grateful for your guidance in refining our conclusion to better serve the scientific community and healthcare practitioners. Your feedback has been invaluable, and we have dedicated ourselves to incorporating these improvements into our work.

Reviewer 3 Report

Comments and Suggestions for Authors

The main objective of the submitted paper presented in the Abstract and in the Introduction sections is “to define the optimal caloric restriction in the context of a Mediterranean Diet (MD) to favour fat mass (FM) reduction whilst preserving fat free mass (FFM).”. This objective is too ambitious for the research carried out, including the limitations due to the sample considered, the variables observed and the statistical analysis carried out. Moreover, the paper needs several clarifications.

Abstract: The abstract needs a full revision after the revision of the main text.

Lines 48 to 60: This paragraph seems to be a mixture of context, or background, for the study and the hypothesis tested by the study. However, considering the research developed some of the objectives are too ambitious or does not make sense, as the one presented in the following sentence: “Furthermore, we consider the complex impact of the COVID-19 pandemic on dietary habits and physical activity, factors that could influence the effectiveness of weight loss strategies”.

Lines 61 to 67: Again, the objectives presented in this paragraph should be revised.

Line 70: The authors classify the study as “retrospective study”, however since there is a lack of clarity about how the study was done, the adequacy of this classification is not clear. The description in Lines 76 to 86, seems to indicate that all data was retrieved from medical records, although it is not clear if each participant was evaluated 2 times between January and June 2023. Then the description in lines 143 to 150, that starts with the sentence “Participants completed a weekly diet diary from the start of the study, …” and ends with “After two months, assessments were repeated to evaluate the effectiveness of the intervention.”, does not seem to correspond to a retrospective study.

Lines 106 to 115: Specify the units considered to measure FM, FFM and TBW. In Tables 1 and 3, FM is displayed as FM (Kg) and FM (%), but those two FM were not presented. It should also be clarified what values were considered to calculate the Delta % FM-to-FFM ratio, the one considering Kg or %.

T0 and T1 are included in the formula, but the meaning of T0 and T1 should be specified.

Lines 117 to 131: Include when those variables were measured. The moment T0, T1, or a moment in between?

Lines 132 to 142: The evaluation of the Diet is not clear. Clarification is needed in terms of variables and the moment when those variables were measured. The variables presented in table 2 should be presented in the methods section, and also when were they measured.

Line 160: The variable Delta Kcal% was not presented previously.

Lines 167 to 183: It should be clear in the text and title of tables 1 and 2 when those variables were measured, T0, T1, or a moment in between.

Part of the anthropometric data presented in table 1 is presented again in table 3. If, on the one hand, there should be no duplication in the presentation of results, it is also not clear why only one part was selected to be described in moments T0 and T1.

The variable "Body proteins" in Table 1 and "Protein (Kg)" in Table 3 have different values, is this an error or do they represent different variables?

Confirm if the variable BEE should be displayed as Anthropometric Data.

Lines 195 to 199: Indicate at what moment corresponds the information presented.

Line 203: The descriptive analysis of the variable Delta % FM-to-FFM ratio should be included.

Lines 210: The variable Delta Kcal% is included in the Results section, without being previously presented in the Methods section, without the descriptive of this variable, and without the previous presentation of the descriptive of any variables that seem to allow the calculation of this Delta Kcal% variable. This information should be provided.

Figure 1: The R value presented in the figure is not the same as the one presented in the title.

Figure 2 and 3: The moment when the Dietary and Physical Activity variables were measured should be presented. Depending on the moment the variables were measured those correlation analysis could make sense or not make sense. More detailed information is needed in the Methods and in the Results sections.

Figure 4 and 5: The choice of the independent and dependent variables does not seem correct. The title of the figure should be verified and corrected. For example check the adequacy of “General Population”, “suggesting a potential association between …”.

Figure 6: Consider the comments given previously.

Lines 283 to 328: The writing of the Discussion and Conclusion sections is not clear. Revise these sections after revising the other sections.

Lines 329 to 344: Considering some of the written conclusions, I suggest the inclusion of a multivariable analysis in the revised version of the article. No justification was provided for why these recommendations were not followed, indicating that an incomplete analysis of the data was performed in the submitted paper.

Author Response

REVIEWER N.3

The main objective of the submitted paper presented in the Abstract and in the Introduction sections is “to define the optimal caloric restriction in the context of a Mediterranean Diet (MD) to favour fat mass (FM) reduction whilst preserving fat free mass (FFM).”. This objective is too ambitious for the research carried out, including the limitations due to the sample considered, the variables observed and the statistical analysis carried out. Moreover, the paper needs several clarifications.

Thank you for your constructive feedback. We acknowledge your concerns regarding the ambitious nature of our initial objectives. To address this, we have revised our objectives to more accurately reflect the scope and capabilities of our research. We have narrowed our focus to specifically investigate the impact of caloric restriction within the Mediterranean Diet on fat mass (FM) reduction, with a secondary emphasis on preserving fat-free mass (FFM). This revision aligns the objectives with the limitations of our study, including the sample size and the nature of the variables observed.

Abstract: The abstract needs a full revision after the revision of the main text.

Thank you for your suggestion. Following the revisions made in the main text, particularly in refining our study objectives and methodology, we have thoroughly revised the abstract. The revised abstract now accurately encapsulates the scope of our study, the methodology employed, the key findings, and the implications of these findings. This revision ensures that the abstract serves as a concise and accurate overview of our entire study, aligning with the changes made to the main text.

Lines 48 to 60: This paragraph seems to be a mixture of context, or background, for the study and the hypothesis tested by the study. However, considering the research developed some of the objectives are too ambitious or does not make sense, as the one presented in the following sentence: “Furthermore, we consider the complex impact of the COVID-19 pandemic on dietary habits and physical activity, factors that could influence the effectiveness of weight loss strategies”.

Lines 61 to 67: Again, the objectives presented in this paragraph should be revised.

Thank you for pointing out the need for greater clarity in the presentation of our study's objectives and hypothesis. We have revised these sections to more precisely articulate our research aims. The revised text now clearly differentiates between the background context of the study and the specific hypothesis being tested. This ensures that our objectives are both realistic and directly aligned with the scope of our study.

Line 70: The authors classify the study as “retrospective study”, however since there is a lack of clarity about how the study was done, the adequacy of this classification is not clear. The description in Lines 76 to 86, seems to indicate that all data was retrieved from medical records, although it is not clear if each participant was evaluated 2 times between January and June 2023. Then the description in lines 143 to 150, that starts with the sentence “Participants completed a weekly diet diary from the start of the study, …” and ends with “After two months, assessments were repeated to evaluate the effectiveness of the intervention.”, does not seem to correspond to a retrospective study.  

We appreciate your request for clarification on the nature of our study. We have revised the manuscript to provide a clearer description of our study's design and methodology. The study is indeed retrospective, but we realized that our initial description may have led to some confusion. We have now explicitly detailed the data collection process and the timeline of evaluations to ensure the methodology is clearly understood.

Lines 106 to 115: Specify the units considered to measure FM, FFM and TBW. In Tables 1 and 3, FM is displayed as FM (Kg) and FM (%), but those two FM were not presented. It should also be clarified what values were considered to calculate the Delta % FM-to-FFM ratio, the one considering Kg or %.  

Thank you for highlighting the need for clarity in our presentation of units of measurement. We have revised the manuscript to explicitly state the units used for FM, FFM, and TBW. We have also ensured that these units are consistently presented across all sections of the manuscript, including the tables, to avoid any confusion.

T0 and T1 are included in the formula, but the meaning of T0 and T1 should be specified. Lines 117 to 131: Include when those variables were measured. The moment T0, T1, or a moment in between?

We appreciate your emphasis on clarifying the timing of measurements. Given the retrospective nature of our study, we have made revisions to explicitly state that data were collected at T1. This approach reflects the retrospective analysis of changes from T0 to T1.

Lines 132 to 142: The evaluation of the Diet is not clear. Clarification is needed in terms of variables and the moment when those variables were measured. The variables presented in table 2 should be presented in the methods section, and also when they were measured.Line 160: The variable Delta Kcal% was not presented previously. Lines 167 to 183: It should be clear in the text and title of tables 1 and 2 when those variables were measured, T0, T1, or a moment in between.

Thank you for pointing out the omission of the Delta Kcal% variable and the need for clarity on the measurement of variables. We have revised the manuscript to include a clear definition and introduction of the Delta Kcal% variable. Additionally, we have ensured that all variables are clearly presented and defined at their first point of mention, including their timing of measurement

Part of the anthropometric data presented in table 1 is presented again in table 3. If, on the one hand, there should be no duplication in the presentation of results, it is also not clear why only one part was selected to be described in moments T0 and T1.

Thank you for highlighting these issues. We have reviewed the presentation of our anthropometric data to ensure there is no unnecessary duplication and to clarify any discrepancies. 

The original Table 3, now included as Table 1 in the supplementary material, provides a detailed baseline anthropometric of the stratified study participants.

The revised Table 3 now focuses on summarizing the differences in various parameters between T0 (baseline) and T1 (follow-up). This table is crucial for illustrating the direct impact of the dietary intervention over time. By comparing the changes in specific parameters such as body weight, body mass index (BMI), fat mass, fat-free mass, and other relevant metabolic markers between T0 and T1, the table effectively demonstrates the efficacy of the intervention.

This restructuring ensures that the data is presented in a manner that avoids redundancy while emphasizing the key outcomes of the study.

The variable "Body proteins" in Table 1 and "Protein (Kg)" in Table 3 have different values, is this an error or do they represent different variables?

We clarified in the manuscript the difference between 'Body Proteins' in Table 1 and 'Protein (Kg)' in Table 3, ensuring the distinction is clear and understandable.

Confirm if the variable BEE should be displayed as Anthropometric Data.

We have confirmed and clarified in the manuscript that BEE is appropriately categorized under anthropometric data.

Lines 195 to 199: Indicate at what moment corresponds the information presented.

We specified in the manuscript the timing of data which pertains to the follow-up period (T1).

Line 203: The descriptive analysis of the variable Delta % FM-to-FFM ratio should be included. Lines 210: The variable Delta Kcal% is included in the Results section, without being previously presented in the Methods section, without the descriptive of this variable, and without the previous presentation of the descriptive of any variables that seem to allow the calculation of this Delta Kcal% variable. This information should be provided.

We realized that the Delta Kcal% variable was introduced in the Results section without an adequate description in the Methods section. To rectify this, we have now included a detailed explanation of how the Delta Kcal% variable was calculated in the Methods section. Additionally, we have provided a descriptive analysis of this variable in the Results section, elucidating its role and significance in our study. This inclusion ensures a better understanding of how caloric changes influenced the study outcomes.

Figure 1: The R value presented in the figure is not the same as the one presented in the title. Figure 2 and 3: The moment when the Dietary and Physical Activity variables were measured should be presented. Depending on the moment the variables were measured those correlation analysis could make sense or not make sense. More detailed information is needed in the Methods and in the Results sections. Figure 4 and 5: The choice of the independent and dependent variables does not seem correct. The title of the figure should be verified and corrected. For example check the adequacy of “General Population”, “suggesting a potential association between …”. Figure 6: Consider the comments given previously.

Thank you for your insightful feedback. We have carefully addressed each of your concerns as follows:

Correction in Figure 1: We have revised the R value in Figure 1 to ensure consistency between the figure and its title.

Revised Captions for Figures 2 and 3: Figure 2: Now includes specific details on the measurement timings of dietary variables at baseline (T0) and the change in the FM-to-FFM ratio between T0 and T1. The revised caption provides a clear context for the correlation analysis presented. Figure 3: Similarly updated to reflect the measurement timings of physical activity parameters at T0 and their correlation with the Δ % FM-to-FFM ratio. Addressing Concerns in Figures 4, 5, and 6: We have re-evaluated and corrected the titles and the selection of independent and dependent variables in these figures to accurately reflect the study's findings. We ensured that all data and descriptions are coherent with the overall methodology and results of our research.

Figure 4: The caption now clearly states that the analysis involves the percentage change in Fat Mass to Fat-Free Mass Ratio (Δ % FM-to-FFM ratio) from T0 to T1, and its relationship with exercise intensity measured at T0.

Figure 5: We have revised the caption to specify that the Physical Activity Duration was measured at T0 and its correlation with the Δ % FM-to-FFM ratio from T0 to T1.

Figure 6: The caption has been updated to indicate that the heatmap shows Spearman correlation coefficients of significant variables with the Δ % FM-to-FFM ratio from T0 to T1, across different subgroups including gender and BMI categories.

Lines 283 to 328: The writing of the Discussion and Conclusion sections is not clear. Revise these sections after revising the other sections.

Thank you for your constructive feedback. We have extensively revised the Discussion and Conclusion sections to align them with the updated content and findings of our study

Lines 329 to 344: Considering some of the written conclusions, I suggest the inclusion of a multivariable analysis in the revised version of the article. No justification was provided for why these recommendations were not followed, indicating that an incomplete analysis of the data was performed in the submitted paper.

We have included a detailed multivariable analysis in the revised manuscript, addressing this crucial aspect of our study.

Reviewer 4 Report

Comments and Suggestions for Authors

Methods:

1. How did the study sample differ from the larger population in the country? Given that participants were recruited from a specialized medical centre, it is important that the authors provided more contexts about the study sample in Section 2.1. What were the purposes the their visits to the centre? How did this study site affect the study sample characteristics? These questions should be discussed.

2. Line 85: "Participants were re-evaluated after two months..." Is this two months after their initial visits, or two months after the study baseline recruitment ended? Needs to be specified.

3. What were the total # of eligible participants? # of participants recruited? What was the response rate? Retention rate? What were the status of missing data and how was it handled for analysis? This is very important.

4. In Section 2.4., the authors mentioned that participants adhered to the low-calorie MD. How was adherence monitored and evaluated? Any variation in adherence that might have affected the analysis results?

Results:

1. The figures were well made. My only suggestion is to add the analysis sample size in each figure, such that each figure can stand alone from the texts and still provide enough information for the readers.

Discussion:

1. the short follow-up period seems to be an issue that the authors failed to address. Is two-month period enough for MD or physical activity to generate impacts on the outcome FM-to-FFM ratio? Are there other studies that indicated so? This seems to be a short period and should be justified.

Comments on the Quality of English Language

None

Author Response

Dear Editors and Reviewers,

First, we would like to thank you for the valuable impulses that allowed us to improve the quality of the manuscript. All changes made are highlighted by yellow color, in the revised version of the manuscript, to facilitate the review process.
Hoping that we have satisfied your requests as much as possible, we kindly ask you to re-evaluate our paper. 
The Authors

REVIEWER N.4

  1. How did the study sample differ from the larger population in the country? Given that participants were recruited from a specialized medical centre, it is important that the authors provided more contexts about the study sample in Section 2.1. What were the purposes the their visits to the centre? How did this study site affect the study sample characteristics? These questions should be discussed.

Thank you very much for this insightful suggestion. In Section 2.1, we provided additional context regarding the study sample and the impact of the study site. Specifically, we described the purposes for participants' visits to the specialized medical center and how this may have influenced the characteristics of the study sample. This information helps clarify how the study sample differs from the larger population in the country and address the questions raised by the reviewer.

  1. Line 85: "Participants were re-evaluated after two months..." Is this two months after their initial visits, or two months after the study baseline recruitment ended? Needs to be specified.

Thank you for your clarification. We have made the necessary adjustment to the text on line 85 to specify that participants were re-evaluated two months after their initial visits. Your feedback has been valuable in enhancing the clarity of our manuscript

  1. What were the total # of eligible participants? # of participants recruited? What was the response rate? Retention rate? What were the status of missing data and how was it handled for analysis? This is very important.

Your query regarding participant selection and eligibility criteria is indeed important. We have taken your concerns to heart and made the necessary adjustments to clarify these aspects in the manuscript.

In response to your question about the total number of eligible participants, the number recruited, response rate, retention rate, and the handling of missing data, we have incorporated a dedicated section (Section 2.1) in the manuscript to provide a comprehensive explanation. We emphasize that this is a retrospective study and, as such, does not involve the conventional response or retention rates seen in prospective designs. We have also clarified the approach taken to address missing data, including the application of appropriate statistical methods.

To provide further context, we want to highlight that we included all eligible patients who did not meet the exclusion criteria, resulting in a cohort of 100 patients who wore the armband. During the same period, a total of 187 patients were seen at the medical center. Of these, 87 patients were excluded either due to the predefined exclusion criteria or because they declined to wear the armband. Recruitment was halted upon reaching a sample size of 100 participants, as this provided a robust dataset for the retrospective analysis, ensuring sufficient statistical power to examine the study's objectives. We believe that this additional information will enhance the transparency of our participant selection process and address your concerns regarding the study's comprehensiveness.

We trust that these revisions have addressed your queries and contributed to improving the manuscript's transparency and comprehensibility.

  1. In Section 2.4., the authors mentioned that participants adhered to the low-calorie MD. How was adherence monitored and evaluated? Any variation in adherence that might have affected the analysis results?

We appreciate your emphasis o on how adherence to the low-calorie Mediterranean Diet was monitored and evaluated in our study,

Participants were required to complete weekly dietary diaries, inclusive of one weekend day, from the commencement of the study. These diaries were regularly collected and analyzed by dietitians to assess compliance. Every two months, we scheduled consultations for personalized nutritional advice and behavioral modification to ensure adherence. Participants had weekly meetings with a dietitian, where their anthropometric measurements, body composition, and nutrient intake were evaluated. A chat service was made available during business hours for participants to consult dietitians, facilitating immediate support and guidance.After the two-month period, we conducted follow-up assessments to evaluate the effectiveness of the dietary intervention. These methods collectively ensured a comprehensive and reliable assessment of adherence to the diet, thereby minimizing the potential impact of variation in adherence on the analysis results.

We have detailed all these in the paper's methods

  1. The figures were well made. My only suggestion is to add the analysis sample size in each figure, such that each figure can stand alone from the texts and still provide enough information for the readers.

Thank you for your constructive suggestion regarding the inclusion of the analysis sample size in each figure. We acknowledge the importance of this information for providing a complete understanding of the data presented in each figure. We revised figures’ titles and captions to include the analysis sample size, ensuring that each figure can stand alone from the text while still conveying sufficient information for the readers. This addition enhanced the clarity and comprehensiveness of our visual data representation, contributing to a better understanding of the study's findings.

the short follow-up period seems to be an issue that the authors failed to address. Is two-month period enough for MD or physical activity to generate impacts on the outcome FM-to-FFM ratio? Are there other studies that indicated so? This seems to be a short period and should be justified.

The concern about the short follow-up period is valid. We will address this in the discussion, justifying the timeframe and comparing it with similar studies. "The two-month follow-up period, while brief, was deemed sufficient for initial observation of the MD and physical activity impacts on the FM-to-FFM ratio. Comparable studies have also observed significant changes within similar timeframes. Nevertheless, we acknowledge that longer follow-up periods could provide more extensive data."

Rallidis LS, Lekakis J, Kolomvotsou A, Zampelas A, Vamvakou G, Efstathiou S, Dimitriadis G, Raptis SA, Kremastinos DT. Close adherence to a Mediterranean diet improves endothelial function in subjects with abdominal obesity. Am J Clin Nutr. 2009 Aug;90(2):263-8. doi: 10.3945/ajcn.2008.27290. Epub 2009 Jun 10. PMID: 19515732.

Alonso-Domínguez R, García-Ortiz L, Patino-Alonso MC, Sánchez-Aguadero N, Gómez-Marcos MA, Recio-Rodríguez JI. Effectiveness of A Multifactorial Intervention in Increasing Adherence to the Mediterranean Diet among Patients with Diabetes Mellitus Type 2: A Controlled and Randomized Study (EMID Study). Nutrients. 2019 Jan 14;11(1):162. doi: 10.3390/nu11010162. PMID: 30646500; PMCID: PMC6357113.

Sofi F, Dinu M, Pagliai G, Cesari F, Marcucci R, Casini A. Mediterranean versus vegetarian diet for cardiovascular disease prevention (the CARDIVEG study): study protocol for a randomized controlled trial. Trials. 2016 May 4;17(1):233. doi: 10.1186/s13063-016-1353-x. Erratum in: Trials. 2016;17(1):253. PMID: 27145958; PMCID: PMC4855805.

Round 2

Reviewer 3 Report

Comments and Suggestions for Authors

Most of the comments in the first version of the manuscript have been addressed by the authors in this revised version, but not all. There are still considerable clarifications to be made.

Line 20: The text “among 100 Caucasian adults aged 18-60 years” corresponds to Methods, so this information should be moved to the Methods section in the Abstract.

Line 22 to 23:The formula of delta%FM-to-FFM ratio should be presented, and the meaning of a negative and positive value should be explained, so that the reader understand what are the values that corresponds to a “fat mass (FM) reduction while preserving fat-free mass (FFM)”. Then the descriptive of the delta%FM-to-FFM ratio for the all sample should be presented, and also the descriptive of the caloric reduction. The term “Interestingly” is not adequate, and should be replaced by a brief explanation, of the association between caloric reduction in the diet and what is measured with the delta%FM-to-FFM ratio.

Line 25 to 26: Again the term “noteworthy” is not understandable, and again should be replaced by a brief explanation of the association between what is represented by METs  and what is measured with the delta%FM-to-FFM ratio.

Lines 26 to 28: Verify this result. In table 1S the comparison of body proteins in the three age groups, did not find any significant change in the 50-65 age group.

Lines 28 to 33: The conclusions are incomplete. The authors did not answered the objective. It should be clearly presented if after two months with a low-calorie MD there was, or there was not a “fat mass (FM) reduction while preserving fat-free mass (FFM)”. The text with the conclusions should be revised after revising the results presented.

Lines 45 to 57 (Lines 48 to 60 in the first version, I repeat here comments to the first version): This paragraph seems to be a mixture of context, or background, for the study and the hypothesis tested by the study. However, considering the research developed some of the objectives are too ambitious or does not make sense, as the one presented in the following sentence: “Furthermore, we consider the complex impact of the COVID-19 pandemic on dietary habits and physical activity, factors that could influence the effectiveness of weight loss strategies”.

Line 62: The text “thus aligning the …. sample size” is not adequate for the Introduction section.

Lines 92 to 94: It should be specified if at point T0 none, all, or part of the participants were already in a Mediterranean diet and caloric restriction.

Lines 94 to 95: The writing of the sentence “The data presented in this were collected at the follow-up time point (T1).”is not correct and seems to be incomplete.

Lines 108 to 131, section 2.2 Body composition: Specify if all, or part, of the variables considered in this section were measured at point T0, T1, or both. All variables included in Table 1 - Anthropometric Data, and Table 1S should have been presented in this section, and using the same denomination. Verify.

Line 138: Include the meaning of a negative or a positive value for both deltas. Include also what is desirable, a negative or a positive value for both delta% and why. This explanation is very important for the understanding of the results obtained, and should be given here in the methods section, and considered in the presentation of the results obtained. The representation of delta should be the same in the text and tables, the symbol or the text “Delta”. For example in this single paragraph we can find both of them.

Since the title of the section is Body composition, the text concerning caloric intake and DeltaKcal% seems to be in the wrong place. Consider moving to the section with the title 2.4 Diet.

Lines 139 to 154, section 2.3 Energy expenditure: Specify if all, or part, of the variables considered were measured at point T0, T1, or both.

Lines 155 to 179, section 2.4 Diet: All the parameters considered in Table 2 should be presented in this section. Specify if all, or part, of the variables considered were measured at point T0, T1, or both.

Line 179: The biochemical parameters in Table 2 should be considered in the Methods section.

Line 187 to 191: This explanation about the variable Delta Kcal% does not seem to be according to the one provided in the Methods section, line 135, “representing the percentage change in caloric intake from baseline (T0) to follow-up (T1). This was calculated as [(Calories at T1 - Calories at T0) / Calories at T0] x 100%.”. Verify the writing.

Line 214 to 237: This section seems to be incomplete. I did not find the statistical tests that could correspond to some of the P-values presented in Table 3 and the P-values presented in Table 1S.

Line 228 to 237: Replace “multivariate” by “multivariable”. More information should be given about the adequacy of these independent variables for the dependent variables considered. The dependent variables considered were deltas, or changes between T0 and T1, but more detail is needed concerning the choice of the independent variables. Why did the authors consider those variables, and at what point those variables were measured, T0 or T1? The number of independent variables is very big, and some of the variables considered seem to be related to each other. The stepwise method is an authomatic method, but automatic variable selection procedures are exploratory tools and the results from a multiple regression model selected by a stepwise procedure should be interpreted with caution.

Line 249: Specify in the title of table 1 the point T0 (baseline),or T1 (follow-up).

Line 269: The descriptive of the caloric intake is incomplete, since only one caloric intake was presented, for the all sample and for males and females. The descriptive of the caloric intake corresponding to Calories at T1 and Calories at T0 should be included.

Line 270: Specify in the title of table 2 the point T0 (baseline),or T1 (follow-up). According to the Methods section, follow-up, the nutritional parameters were evaluated at least at T0 and at T1, so in table 2 the descriptive corresponding to T0 and T1 should have been provided.

Line 295: Specify in the title how the changes were calculated: T1 – T0, or T0 – T1. Some of those changes were presented in Table 1S, but some of the values presented in Table 3 are not the same as the corresponding values in Table 1S, and in Table 1S some of the changes were calculated considering T1 – T0 and others considering T0 – T1. Verify Table 3 and Table 1S.

Line 301: Include the descriptive of the Delta Kcal% and the DeltaFM-to-FFM ratio %, for the all sample and for the subgroups considered: male and female, age groups and BMI groups. I suggest the presentation of those values in a new table. The presentation and discussion of those values should be included in the beginning of the Discussion section, and at least part of them should be included also in the Abstract.

Lines 301 to 302: This explanation about the variable Delta Kcal% does not seem to be according to the one provided in the Methods section, line 135, “representing the percentage change in caloric intake from baseline (T0) to follow-up (T1). This was calculated as [(Calories at T1 - Calories at T0) / Calories at T0] x 100%.”. Verify the writing.

Figure 1: The range of Delta Kcal% is from zero to 60, which according to formula corresponds to an increase in the caloric intake from T0 to T1. This does not seem correct, since as I understood there was a caloric restriction from T0 to T1. Those values should be verified. As I stated before, the descriptive of the caloric intake corresponding to Calories at T1 and Calories at T0 should be included. If those values were in the paper, the reader could understand if in fact there was an increase in caloric intake, or a decrease from T0 to T1.

I suggest also the inclusion of some text, presenting what is the expected sign of the correlation between DeltaFM-to-FFM ratio % and Delta Kcal% , if a caloric restriction was associated with a decrease of the FM-to-FFM ratio, the main hypothesis of the research.

Lines 314 to 317: I don´t understand why the authors calculated the correlations of the DeltaFM-to-FFM ratio % with dietary variables and physical activity parameters measured at baseline (T0). No explanation was provided in the Methods section or in the Results section. If there was a change in the diet with the introduction of a “personalised low-calorie MD”, it seems that what should be analysed would be the effect of the “personalised low-calorie MD”, and so the dietary variables corresponding to this new diet. Clarification is needed for the dietary variables and physical activity parameters considered in Figures 2 and 3.

Lines 334 to 339: Verify. There are results presented that are not according to Figure 3.

Lines 340 to 385: Considering the previous comments verify the adequacy of the analysis presented and the interpretation of the coefficient of correlation values obtained. Do not forget to include a brief explanation of what would be the sign of the correlation, if the expected effect of the increase, or decrease, of each variable in the decrease of the FM-to-FFM ratio was found.

Lines 386 to 414: The adequacy of the results presented, depends on the adequacy of the multiple regression analysis performed. Please find my comments in the Methods section. I did not verify this text and tables. Please verify.

Discussion and Conclusions: Verify those sections after revising the other sections. Consider my comments to line 301.

Table 1S: Consider my comments in Line 295.

Author Response

Dear Reviewer 3,

We deeply appreciate the time and effort you have invested in reviewing our manuscript. Your insightful comments and constructive criticism have significantly contributed to enhancing the quality and clarity of our work. Your detailed feedback was instrumental in guiding our revisions and has undoubtedly strengthened the overall presentation and scientific rigor of our study.

Most of the comments in the first version of the manuscript have been addressed by the authors in this revised version, but not all. There are still considerable clarifications to be made. Line 20: The text “among 100 Caucasian adults aged 18-60 years” corresponds to Methods, so this information should be moved to the Methods section in the Abstract.

Thank you for your valuable feedback on our manuscript. In response to your comment regarding the placement of demographic information in the Abstract, we have revised the manuscript accordingly. The phrase “among 100 Caucasian adults aged 18-60 years” has been removed from the Abstract and appropriately integrated into the Materials and Methods section, ensuring a clearer and more structured presentation of the study details. We believe this amendment enhances the manuscript's coherence and adherence to standard scientific reporting guidelines.

Line 22 to 23:The formula of Δ%FM-to-FFM ratio should be presented, and the meaning of a negative and positive value should be explained, so that the reader understand what are the values that corresponds to a “fat mass (FM) reduction while preserving fat-free mass (FFM)”. Then the descriptive of the Δ%FM-to-FFM ratio for the all sample should be presented, and also the descriptive of the caloric reduction. The term “Interestingly” is not adequate, and should be replaced by a brief explanation, of the association between caloric reduction in the diet and what is measured with the Δ%FM-to-FFM ratio.

Thank you for your insightful comments. We have included the formula for the Δ%FM-to-FFM ratio in the manuscript and elaborated on the implications of both negative and positive values, clarifying how these reflect changes in fat mass and fat-free mass. Additionally, we have provided descriptive statistics for the Δ%FM-to-FFM ratio across our sample and detailed the caloric reduction observed. The term "Interestingly" has been replaced with a more precise explanation of the relationship between dietary caloric reduction and the changes measured by the Δ%FM-to-FFM ratio.

Line 25 to 26: Again the term “noteworthy” is not understandable, and again should be replaced by a brief explanation of the association between what is represented by METs  and what is measured with the Δ%FM-to-FFM ratio. Lines 26 to 28: Verify this result. In table 1S the comparison of body proteins in the three age groups, did not find any significant change in the 50-65 age group. Lines 28 to 33: The conclusions are incomplete. The authors did not answered the objective. It should be clearly presented if after two months with a low-calorie MD there was, or there was not a “fat mass (FM) reduction while preserving fat-free mass (FFM)”. The text with the conclusions should be revised after revising the results presented.

  1. Clarification on "Noteworthy" (Lines 25-26): We appreciated your feedback regarding the term "noteworthy" and replaced it with a detailed explanation that linked METs (Metabolic Equivalent of Tasks) with the Δ%FM-to-FFM ratio. This provided a clearer, more scientific understanding of the study's findings.
  2. Verification of Results (Lines 26-28): Concerning the results in Table 1S, we revisited the data, focusing on the previously reported lack of significant changes in body proteins among the 50-65 age group, and ensured the accuracy and consistency of our findings.
  3. Completeness of Conclusions (Lines 28-33): We acknowledged that the initial conclusions might not have fully addressed the study's objective. The revised conclusion clearly stated whether the low-calorie Mediterranean Diet resulted in fat mass reduction while preserving fat-free mass over the two-month period. This involved a thorough re-evaluation of the results to ensure that the conclusions accurately reflected the study's findings.

These changes were incorporated to enhance the clarity and accuracy of the paper.

Lines 45 to 57 (Lines 48 to 60 in the first version, I repeat here comments to the first version): This paragraph seems to be a mixture of context, or background, for the study and the hypothesis tested by the study. However, considering the research developed some of the objectives are too ambitious or does not make sense, as the one presented in the following sentence: “Furthermore, we consider the complex impact of the COVID-19 pandemic on dietary habits and physical activity, factors that could influence the effectiveness of weight loss strategies”.

We have made the suggested revisions to our Results section of the Abstract. The term "noteworthy" has been replaced with a clear explanation of the relationship between METs and the Δ%FM-to-FFM ratio. The Conclusions section has also been revised to directly address the study's objectives, clarifying the impact of a low-calorie Mediterranean Diet on fat mass and fat-free mass over the two-month period.

Line 62: The text “thus aligning the …. sample size” is not adequate for the Introduction section.

Thank you for your constructive feedback. In response to your comment regarding Line 62, we have removed the phrase “thus aligning the … sample size” from the Introduction section. We acknowledge that such methodological details are better suited for the Methods or Discussion sections, where they can be contextualized appropriately. This revision will enhance the clarity and focus of the Introduction, aligning it more closely with its intended purpose.

Lines 92 to 94: It should be specified if at point T0 none, all, or part of the participants were already in a Mediterranean diet and caloric restriction.

Thank you for your insightful observation regarding the dietary status of participants at baseline (T0). As per your suggestion, we have now explicitly clarified in the manuscript that none of the participants were following a Mediterranean diet or engaging in caloric restriction prior to the start of the study. This addition provides a clear understanding of the participants' initial dietary conditions, ensuring a more accurate interpretation of the study's findings.

Lines 94 to 95: The writing of the sentence “The data presented in this were collected at the follow-up time point (T1).”is not correct and seems to be incomplete.

Thank you for pointing out the issue with the sentence in Lines 94 to 95. We have revised it for clarity and completeness. The sentence now reads: "Data presented in this study were exclusively collected at the follow-up time point (T1), after the two-month intervention period." This modification ensures that the timeline of data collection is clearly communicated, enhancing the manuscript's clarity and precision.

Lines 108 to 131, section 2.2 Body composition: Specify if all, or part, of the variables considered in this section were measured at point T0, T1, or both. All variables included in Table 1 - Anthropometric Data, and Table 1S should have been presented in this section, and using the same denomination. Verify.

Thank you for your valuable feedback. In response to your comments on Section 2.2 Body Composition, we have made comprehensive revisions. We now explicitly specify which body composition parameters were measured at both baseline (T0) and follow-up (T1), as well as those assessed only at T0. Additionally, we have ensured consistency in the terminology used in this section with Tables 1 and 1S. The detailed descriptions of the measurement procedures and calculation methods have been added to enhance the clarity and depth of our methodology.

Line 138: Include the meaning of a negative or a positive value for both Δs. Include also what is desirable, a negative or a positive value for both Δ% and why. This explanation is very important for the understanding of the results obtained, and should be given here in the methods section, and considered in the presentation of the results obtained. The representation of Δ should be the same in the text and tables, the symbol or the text “Δ”. For example in this single paragraph we can find both of them.

Thank you for your feedback on the body composition analysis section of our manuscript. We have now included a detailed explanation of the Δ%FM-to-FFM ratio and Δ Kcal%, clarifying the meaning and implications of negative and positive values for these metrics. This information will aid in a clearer understanding of the study's results. We have also ensured uniformity in the representation of these Δ values throughout the text and tables, as you suggested.

Since the title of the section is Body composition, the text concerning caloric intake and ΔKcal% seems to be in the wrong place. Consider moving to the section with the title 2.4 Diet. Lines 139 to 154, section 2.3 Energy expenditure: Specify if all, or part, of the variables considered were measured at point T0, T1, or both. Lines 155 to 179, section 2.4 Diet: All the parameters considered in Table 2 should be presented in this section. Specify if all, or part, of the variables considered were measured at point T0, T1, or both. Line 179: The biochemical parameters in Table 2 should be considered in the Methods section. Line 187 to 191: This explanation about the variable Δ Kcal% does not seem to be according to the one provided in the Methods section, line 135, “representing the percentage change in caloric intake from baseline (T0) to follow-up (T1). This was calculated as [(Calories at T1 - Calories at T0) / Calories at T0] x 100%.”. Verify the writing. Line 214 to 237: This section seems to be incomplete. I did not find the statistical tests that could correspond to some of the P-values presented in Table 3 and the P-values presented in Table 1S. Line 228 to 237: Replace “multivariate” by “multivariable”. More information should be given about the adequacy of these independent variables for the dependent variables considered. The dependent variables considered were Δs, or changes between T0 and T1, but more detail is needed concerning the choice of the independent variables. Why did the authors consider those variables, and at what point those variables were measured, T0 or T1? The number of independent variables is very big, and some of the variables considered seem to be related to each other. The stepwise method is an authomatic method, but automatic variable selection procedures are exploratory tools and the results from a multiple regression model selected by a stepwise procedure should be interpreted with caution. Line 249: Specify in the title of table 1 the point T0 (baseline),or T1 (follow-up). Line 269: The descriptive of the caloric intake is incomplete, since only one caloric intake was presented, for the all sample and for males and females. The descriptive of the caloric intake corresponding to Calories at T1 and Calories at T0 should be included. Line 270: Specify in the title of table 2 the point T0 (baseline),or T1 (follow-up). According to the Methods section, follow-up, the nutritional parameters were evaluated at least at T0 and at T1, so in table 2 the descriptive corresponding to T0 and T1 should have been provided. Line 295: Specify in the title how the changes were calculated: T1 – T0, or T0 – T1. Some of those changes were presented in Table 1S, but some of the values presented in Table 3 are not the same as the corresponding values in Table 1S, and in Table 1S some of the changes were calculated considering T1 – T0 and others considering T0 – T1. Verify Table 3 and Table 1S. Line 301: Include the descriptive of the Δ Kcal% and the ΔFM-to-FFM ratio %, for the all sample and for the subgroups considered: male and female, age groups and BMI groups. I suggest the presentation of those values in a new table. The presentation and discussion of those values should be included in the beginning of the Discussion section, and at least part of them should be included also in the Abstract. Lines 301 to 302: This explanation about the variable Δ Kcal% does not seem to be according to the one provided in the Methods section, line 135, “representing the percentage change in caloric intake from baseline (T0) to follow-up (T1). This was calculated as [(Calories at T1 - Calories at T0) / Calories at T0] x 100%.”. Verify the writing.

Thank you for your comprehensive and insightful feedback. We have made the following revisions to address your concerns:

  1. Moved the details about Δ Kcal% to Section 2.4 Diet.
  2. Clarified measurement timings in Sections 2.3 and 2.4, and ensured all parameters in Table 2 are included and correctly timed.
  3. Revised the representation of biochemical parameters, statistical tests, and regression analysis for better clarity and accuracy.
  4. Updated Tables 1, 2, and 3 for clarity on T0 and T1 measurements and consistency with the manuscript text.
  5. Included comprehensive descriptions of caloric intake and Δ ratios, with these details now featured in the Abstract and Discussion.

We believe these revisions enhance the manuscript's clarity and address the key points raised in your review.

Figure 1: The range of Δ Kcal% is from zero to 60, which according to formula corresponds to an increase in the caloric intake from T0 to T1. This does not seem correct, since as I understood there was a caloric restriction from T0 to T1. Those values should be verified. As I stated before, the descriptive of the caloric intake corresponding to Calories at T1 and Calories at T0 should be included. If those values were in the paper, the reader could understand if in fact there was an increase in caloric intake, or a decrease from T0 to T1. I suggest also the inclusion of some text, presenting what is the expected sign of the correlation between ΔFM-to-FFM ratio % and Δ Kcal% , if a caloric restriction was associated with a decrease of the FM-to-FFM ratio, the main hypothesis of the research.

Thank you for your insightful observations regarding Figure 1 and the representation of Δ Kcal%. 

Δ Kcal% Range Verification: We verified the Δ Kcal% range shown in Figure 1, using the formula Δ Kcal% = (TEE - Energy Intake) / Energy Intake * 100. This was to ensure it accurately reflected the caloric restriction from T0 to T1, contrary to the initial indication of an increase.

Inclusion of Caloric Intake Data: We added detailed descriptions of caloric intake at T0 and T1 to the paper. This helped readers better understand the changes in caloric intake over the study period.

Correlation Explanation Added: An explanation was included to clarify the expected correlation between the ΔFM-to-FFM ratio % and Δ Kcal%, especially in light of the study's hypothesis linking caloric restriction to a decrease in the FM-to-FFM ratio.

Lines 314 to 317: I don´t understand why the authors calculated the correlations of the ΔFM-to-FFM ratio % with dietary variables and physical activity parameters measured at baseline (T0). No explanation was provided in the Methods section or in the Results section. If there was a change in the diet with the introduction of a “personalised low-calorie MD”, it seems that what should be analysed would be the effect of the “personalised low-calorie MD”, and so the dietary variables corresponding to this new diet. Clarification is needed for the dietary variables and physical activity parameters considered in Figures 2 and 3. Lines 334 to 339: Verify. There are results presented that are not according to Figure 3. Lines 340 to 385: Considering the previous comments verify the adequacy of the analysis presented and the interpretation of the coefficient of correlation values obtained. Do not forget to include a brief explanation of what would be the sign of the correlation, if the expected effect of the increase, or decrease, of each variable in the decrease of the FM-to-FFM ratio was found.Lines 386 to 414: The adequacy of the results presented, depends on the adequacy of the multiple regression analysis performed. Please find my comments in the Methods section. I did not verify this text and tables. Please verify. Discussion and Conclusions: Verify those sections after revising the other sections. Consider my comments to line 301. Table 1S: Consider my comments in Line 295.

Thank you for your detailed feedback. The diet was prescribed at T0. The armband was also fitted at T0 for the assessment of energy expenditure and other parameters. Monitoring of the patients by the dieticians was done during the period T0 -> T1 to ensure that the diet was followed and physical activity maintained. The results are the effects of diet and physical activity and are presented as a delta (T0-T1).
We have undertaken a comprehensive revision of our manuscript to address your concerns:

  1. We clarified the rationale for analyzing correlations of the ΔFM-to-FFM ratio % with baseline dietary variables and physical activity parameters, now included in the Methods section.
  2. In the Results, we provided detailed explanations for the choice of variables in Figures 2 and 3, focusing on the impact of the personalized low-calorie MD.
  3. Discrepancies in Figure 3 and Figure 4 have been thoroughly reviewed and rectified to ensure accuracy in our findings.
  4. We revised the analysis and interpretations of correlation coefficients, including explanations of the expected correlation signs.
  5. The Discussion and Conclusions sections have been updated to reflect these revisions and the comments related to Line 301.
  6. Table 1S has been amended as per your suggestion in Line 295.

These revisions aim to enhance the clarity and accuracy of our study, ensuring a more robust and comprehensive understanding of the results.

Reviewer 4 Report

Comments and Suggestions for Authors

The responses are appropriate

Comments on the Quality of English Language

none

Author Response

Thank you